# Language models scale reliably with over-training and on downstream tasks

## Abstract

Scaling laws are useful guides for derisking expensive training runs, as they predict performance of large models using cheaper, small-scale experiments. However, there remain gaps between current scaling studies and how language models are ultimately trained and evaluated. For instance, scaling is usually studied in the compute-optimal training regime (i.e., "Chinchilla optimal" regime). In contrast, models are often over-trained to reduce inference costs. Moreover, scaling laws mostly predict loss on next-token prediction, but models are usually compared on downstream task performance. To address both shortcomings, we create a testbed of 104 models with 0.011B to 6.9B parameters trained with various numbers of tokens on three data distributions. First, we fit scaling laws that extrapolate in both the amount of over-training and the number of model parameters. This enables us to predict the validation loss of a 1.4B parameter, 900B token run (i.e., $32\times$ over-trained) and a 6.9B parameter, 138B token run (i.e., a compute-optimal run)—each from experiments that take $300\times$ less compute. Second, we relate the perplexity of a language model to its downstream task performance by proposing a power law. We use this law to predict top-1 error averaged over downstream tasks for the two aforementioned models, using experiments that take $20\times$ less compute.

## 1 Introduction

Training large language models is expensive. Furthermore, training high-quality models requires a complex recipe of algorithmic techniques and training data. To reduce the cost of finding successful training recipes, researchers first evaluate ideas with small experiments and then extrapolate their efficacy to larger model and data regimes via scaling laws. With reliable extrapolation, it is possible to quickly iterate at small scale and still pick the method that will perform best for the final large training run. Indeed, this workflow has become commonplace for training state-of-the-art language models like Chinchilla 70B [45], PaLM 540B [19], GPT-4 [76], and many others.

Despite their importance for model development, published scaling laws differ from the goals of training state-of-the-art models in important ways. For instance, scaling studies usually focus on the compute-optimal training regime ("Chinchilla optimality" [45]), where model and dataset size are set to yield minimum loss for a given compute budget. However, this setting ignores inference costs. As larger models are more expensive at inference, it is now common practice to over-train smaller models [113]. Another potential mismatch is that most scaling laws quantify model performance by perplexity in next-token prediction instead of accuracy on widely used benchmark datasets. However, practitioners usually turn to benchmark performance, not loss, to compare models.

In this paper, we conduct an extensive set of experiments to address both scaling in the over-trained regime and benchmark performance prediction.

Submitted to 38th Conference on Neural Information Processing Systems (NeurIPS 2024). Do not distribute.

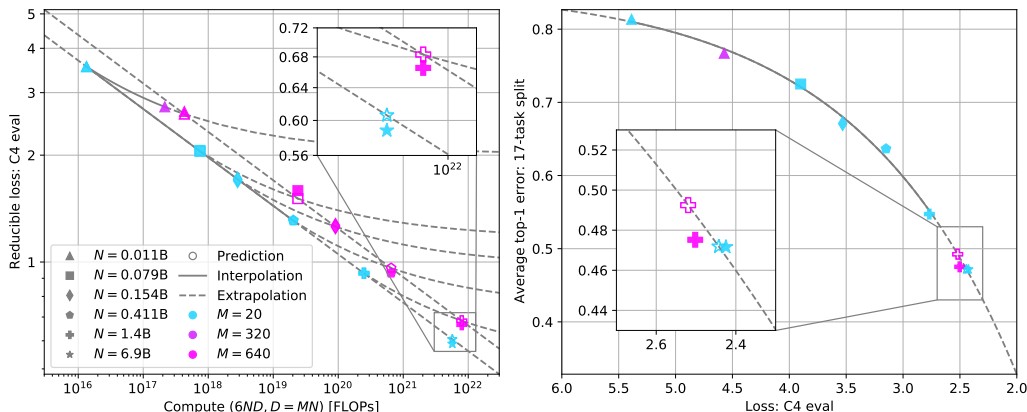

Figure 1: **Reliable scaling with over-training and on downstream error prediction.** *(left)* We fit a scaling law for model validation loss, parameterized by (i) a token multiplier $M = N/D$, which is the ratio of training tokens $D$ to parameters $N$ and (ii) the compute $C$ in FLOPs used to train a model, approximated by $C = 6ND$. Larger values of $M$ specify more over-training. We are able to extrapolate, in both $N$ and $M$, the validation performance of models requiring more than $300\times$ the training compute used to construct the scaling law. *(right)* We also fit a scaling law to predict average downstream top-1 error as a function of validation loss. We find that fitting scaling laws for downstream error benefits from using more expensive models when compared to fitting for loss prediction. We predict the average error over 17 downstream tasks for models trained with over $20\times$ the compute. For this figure, we train all models on RedPajama [112].

Motivated by the practice of training beyond compute-optimality, we first investigate whether scaling follows reliable trends in the over-trained regime. We notice, as implied by Hoffmann et al. [45], for a set of models of different sizes trained with a constant ratio of tokens to parameters, models' reducible loss $L'$ [43, 45] follows a power law ($L' = \lambda \cdot C^{-\eta}$) in the amount of training compute $C$. We find that as one increases the ratio of tokens to parameters, corresponding to more over-training, the scaling exponent $\eta$ remains about the same, while the scalar $\lambda$ changes. We explain our observations by reparameterizing existing scaling laws in relation to the amount of over-training.

To establish empirically that scaling *extrapolates* in the over-trained regime, we further experiment with a testbed of 104 models, trained from scratch on three different datasets: C4 [88, 27], RedPajama [112], and RefinedWeb [82]. We find that scaling laws fit to small models can accurately predict the performance of larger models that undergo more over-training. Figure 1 *(left)* illustrates our main over-training result, where we invest $2.4e19$ FLOPs to extrapolate the C4 validation performance of a 1.4B parameter model trained on 900B tokens, which requires $300\times$ more compute to train.

In addition to over-training, we also investigate if scaling laws can predict the performance of a model on downstream tasks. We establish a power law relationship between language modeling perplexity and the average top-1 error on a suite of downstream tasks. While it can be difficult to predict the error on individual tasks, we find it possible to predict aggregate performance from a model's perplexity among models trained on the same training data. Figure 1 *(right)* presents our main downstream error prediction result, where we invest $2.7e20$ FLOPs to predict the average top-1 error over a set of downstream tasks to within 1 percentage point for a 6.9B compute-optimal model, which requires $20\times$ more compute to train.

Our results suggest that the proposed scaling laws are promising to derisk (i) the effects of over-training models and (ii) the downstream performance of scaling up training recipes. To facilitate further research on reliable scaling, we will release all experiments and models.

## 2 Developing scaling laws for over-training and downstream tasks

In this section, we develop scaling laws to predict over-trained and downstream performance. First, we provide key definitions (Section 2.1). We next present a scaling law for over-training drawing on empirical observation and prior work (Section 2.2). To connect loss scaling and downstream error prediction, we observe that average top-1 error decreases exponentially as a function of validation loss,

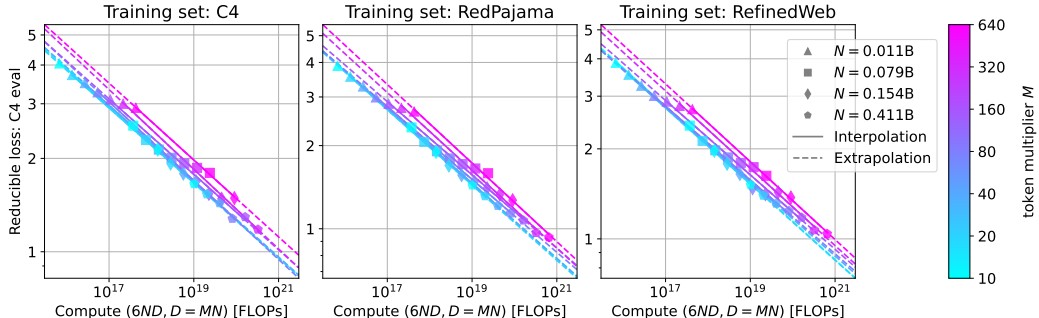

Figure 2: **Scaling in the over-trained regime follows consistent power law exponents.** We notice parallel lines in the $\log$-$\log$ plots of reducible loss vs. training compute for a range of token multipliers $M$, which give the ratio of training tokens to model parameters. Larger $M$ corresponds to more over-training. For a power law giving reducible loss as a function of compute: $L'(C) = \lambda \cdot C^{-\eta}$, the exponent $\eta$ remains relatively constant resulting in lines with approximately fixed slope (Figure 17). The scalar $\lambda$ that determines the $y$-intercept, however, shifts with different token multipliers. This suggests $\lambda$ is a function of the token multiplier, while $\eta$ is not.

which we formalize as a novel scaling law (Section 2.3). In later sections, we build an experimental setup (Section 3) to quantify the extent to which our scaling laws extrapolate reliably (Section 4).

## 2.1 Preliminaries

**Scaling laws for loss.** Typically, scaling laws predict model loss $L$ as a function of the compute $C$ in FLOPs used for training. If one increases the number of parameters $N$ in a model or the number of tokens $D$ that a model is trained on, compute requirements naturally increase. Hence, we assume $C$ is a function of $N, D$. Following Kaplan et al. [51], we use the approximation $C = 6ND$, which Hoffmann et al. [45] independently verify. We consider,

$$L(C) = E + L'(C), \tag{1}$$

where $E$ is an *irreducible loss* and $L'$ is the *reducible loss*. $E$ captures the Bayes error or minimum possible loss achievable on the validation domain. The $L'(C)$ term captures what can possibly be learned about the validation domain by training on a source domain. $L'(C)$ should approach zero with increased training data and model capacity. $L'(C)$ is often assumed to follow a power law: $L'(C) = \lambda \cdot C^{-\eta}$ (i.a., Hestness et al. [43], OpenAI [76]). It is also often helpful to consider a power law in a $\log$-$\log$ plot, where it appears as a line with slope $-\eta$ and $y$-intercept $\log(\lambda)$.

**Token multipliers.** We define a token multiplier $M = D/N$ as the ratio of training tokens to model parameters for notational convenience. $M$ allows us to consider fixed relationships between $D$ and $N$ even as a model gets bigger (i.e., as $N$ becomes larger).

**Compute-optimal training.** Hoffmann et al. [45] establish compute-optimal training, where, for any compute budget $H$, the allocation of parameters and tokens is given by,

$$\arg\min_{N,D} L(N, D) \text{ s.t. } C(N, D) = H. \tag{2}$$

To solve for the optimal $N^*, D^*$, one can sweep $N, D$ for each compute budget, retaining the best configurations. Hoffmann et al. [45] find that as the compute budget increases, $N^*$ and $D^*$ scale roughly evenly. Assuming equal scaling, there is a fixed compute-optimal token multiplier $M^* = D^*/N^*$ per training distribution.

**Over-training.** We define over-training as the practice of allocating compute sub-optimally, so smaller models train on a disproportionately large number of tokens (i.e., $M > M^*$). While loss should be higher than in the compute-optimal allocation for a given training budget, the resulting models have fewer parameters and thus incur less inference cost.

## 2.2 Scaling laws for over-training

To propose a scaling law for over-trained models, we first turn to empirical observation. We train four model configurations with parameter counts between 0.011B and 0.411B for token multipliers $M$

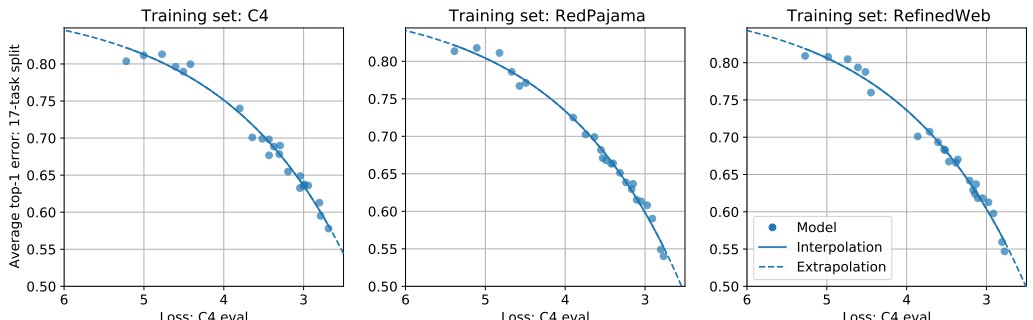

Figure 3: **Average top-1 error scales as a function of loss.** We plot models trained on three datasets and notice an exponential decay of average top-1 error as C4 eval loss, on the x-axis, decreases. We consider on the y-axes average error on 17 evaluations where performance is at least 10 points above random chance for at least one 0.154B scale model. These observations suggest that average top-1 error should be predictable with reliable loss estimates.

between 20 and 640, where $M = 20$ points lie roughly on the compute-optimal frontier, and larger $M$ corresponds to more over-training. We defer experimental details to Section 3 to focus on our observations first. In Figure 2, we show loss against compute in a $\log$-$\log$ plot for the models trained on three datasets and evaluated on the C4 eval set. We notice parallel lines when fitting power laws to the reducible loss, which suggests a near-constant scaling exponent even with increased over-training. This indicates that scaling behavior should be describable in the amount of over-training.

In search of an analytic expression for the observations in Figure 2, we consider existing scaling literature. A common functional form for the risk of a model, as proposed in prior work [93, 45] is,

$$L(N, D) = E + AN^{-\alpha} + BD^{-\beta}. \tag{3}$$

Recall from Section 2.1, $N$ is the number of parameters and $D$ the number of training tokens. The constants $E, A, \alpha, B, \beta$ are fit from data. By fitting this parametric form, Hoffmann et al. [45] find that scaling exponents $\alpha$ and $\beta$ are roughly equal, suggesting that one should scale $N$ and $D$ equally as compute increases. Hence, we assume $\alpha = \beta$. With this assumption, we reparameterize Equation (3) in terms of compute $C = 6ND$ and a token multiplier $M = D/N$. We get,

$$L(C, M) = E + \left(aM^{\eta} + bM^{-\eta}\right) C^{-\eta}, \tag{4}$$

where $\eta = \alpha/2$, $a = A(1/6)^{-\eta}$, $b = B(1/6)^{-\eta}$ gives the relation to Equation (3). For a complete derivation, see Appendix A.

Equation (4) has the following interpretation: (i) The scaling exponent $\eta$ is not dependent on $M$. Thus, we always expect lines with the same slope in the $\log$-$\log$ plot—as in Figure 2. (ii) The term $aM^{\eta} + bM^{-\eta}$ determines the offsets between curves with different token multipliers. Hence, we expect non-overlapping, parallel lines in the $\log$-$\log$ plot for the range of $M$ we consider—also consistent with Figure 2.

Recall that we make the assumption $\alpha = \beta$, which implies equal scaling of parameters and tokens as more compute is available. However, as explained in Appendix A, even if $\alpha \neq \beta$, we get a parameterization that implies the power-law exponent remains constant with over-training.

## 2.3 Scaling laws for downstream error

Scaling is typically studied in the context of loss [51, 45, 72], which Schaeffer et al. [100] note is smoother than metrics like accuracy. However, practitioners often use downstream benchmark accuracy as a proxy for model quality and not loss on perplexity evaluation sets. To better connect scaling laws and over-training to task prediction, we revisit the suite of models plotted in Figure 2. In Figure 3, we plot average downstream top-1 errors over evaluations sourced from LLM-Foundry [69] against the C4 eval loss. We defer details of the setup to Section 3 to focus here on a key observation: average error appears to follow exponential decay as loss decreases.

Based on the exponential decay we observe in Figure 3, we propose the following relationship between downstream average top-1 error Err and loss $L$,

$$\mathsf{Err}(L) = \epsilon - k \cdot \exp\left(-\gamma L\right), \tag{5}$$

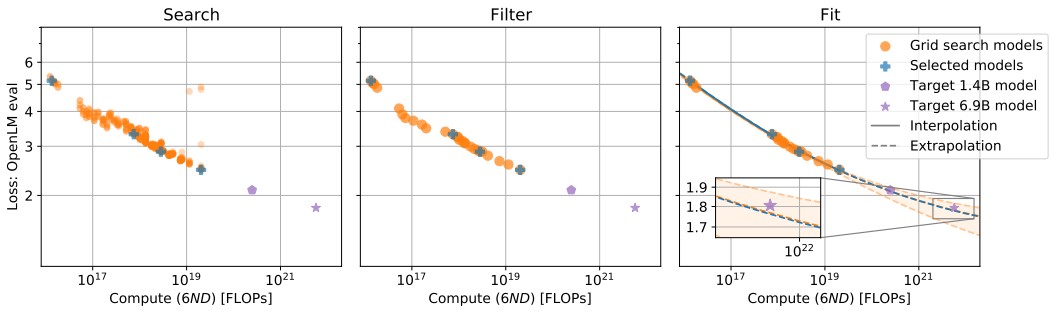

Figure 4: **Search, filter, fit: A recipe for selecting configurations for scaling.** *(left)* To generate the final configurations presented in Table 3, we run a 435 model grid search over model width, hidden dimension, number of attention heads, batch size, and warmup steps. All models are trained near compute-optimally. *(center)* We plot the efficient frontier of models, which appear to follow a trend, excluding models from $5.2 \times 10^{16}$ to $5.2 \times 10^{17}$, which fall below the trend. *(right)* We fit a power law with irreducible error to the remaining configurations, picking four configurations that closely track the full model suite ("Selected models"). These models extrapolate the performance of 1.4B, 6.9B target models. Shaded regions represent bootstrap 95% confidence intervals.

where $\epsilon, k, \gamma$ are fit from data. Equation (5) also has an interpretation in terms of model perplexity $\mathsf{PP}(L) = \exp(L)$,

$$\mathsf{Err}(\mathsf{PP}) = \epsilon - k \cdot \mathsf{PP}^{-\gamma}. \tag{6}$$

Namely, $\mathsf{Err}$ follows a power law in $\mathsf{PP}$ that is bounded from above by $\epsilon$ signifying arbitrarily high error and from below by $\epsilon - k \cdot \exp(-\gamma E)$, where $E$ is the Bayes error from Equation (4).

Equation (5) in conjunction with Equation (4) suggests a three-step method to predict $\mathsf{Err}$ as a function of compute and the amount of over-training. For choices of training and validation distributions, (i) fit a scaling law to Equation (4) using triplets of compute $C$, token multiplier $M$, and measured loss $L$ on a validation set to yield $(C, M) \mapsto L$. (ii) Fit a scaling law to Equation (5) using pairs of loss $L$ and downstream error $\mathsf{Err}$ for models to get $L \mapsto \mathsf{Err}$. (iii) Chain predictions to get $(C, M) \mapsto \mathsf{Err}$.

## 3 Constructing a scaling testbed

In this section, we discuss our experimental setup to test the predictions suggested by Equations (4) and (5). We first present our general language modeling setup (Section 3.1). Next, we discuss our strategy for determining model configurations for our scaling investigation (Section 3.2) and fitting scaling laws (Section 3.3). We then present metrics to validate how well scaling laws predict loss and downstream performance (Section 3.4).

### 3.1 Training setup

We train transformers [116] for next token prediction, based on architectures like GPT-2 [85] and LLaMA [113]. We employ GPT-NeoX [15] as a standardized tokenizer for all data. See Appendix B for architecture, optimization, and hyperparameter details.

### 3.2 Model configurations

To get final configurations for the 0.011B to 0.411B parameter models plotted in Figures 2 and 3, we first conduct a wide grid search over a total of 435 models, trained from scratch, from 0.01B to 0.5B parameters (Figure 4 *(left)*). We train on the original OpenLM data mix [39], which largely consists of RedPajama [112] and The Pile [31]. While we eventually plan to over-train models, at this step we search for *base configurations* near compute-optimality. We train on 20 tokens per parameter ($M = 20$), which, in early experiments, gives models near the compute-optimal frontier. This is similar to findings in Hoffmann et al. [45]'s Table 3, which suggests that $M = 20$ is near-optimal for the Chinchilla experimental setup.

Table 1: **Default number of parameters $N$ and token multiplier $M$ to fit our scaling laws.** We invest $\sim$100 A100 hours to fit Equation (4) and $\sim$1,000 A100 hours to fit Equation (5).

| $N$ | $M$ | Used to fit Equation (4) | Used to fit Equation (5) |
|---|---|---|---|
| 0.011B | 20 | ✓ | ✓ |
| 0.079B | 20 | ✓ | ✓ |
| 0.154B | 20 | ✓ | ✓ |
| 0.411B | 20 | ✓ | ✓ |
| 0.011B | 320 | ✓ | ✓ |
| 1.4B | 20 | ✗ | ✓ |
| Total compute $C$ [FLOPs] | | $2.4e19$ | $2.7e20$ |

To find maximally performant small-scale models on validation data, we tune model width, number of layers, number of attention heads, warmup steps, and batch size. Our validation set, OpenLM eval, contains tokens from recent arXiv papers, the OpenLM codebase itself, and news articles. We find in early experiments that qk-LayerNorm makes models less sensitive to learning rate, which is a phenomenon Wortsman et al. [123] report in their Figure 1. Hence, we fix the learning rate ($3e$-3) for our sweeps. We also perform smaller grid searches over 1.4B and 6.9B parameter model configurations at $M = 20$, retaining the best configurations.

At this point, we have many models, several of which give poor performance; following prior work [51, 45], we want to keep only models that give best performance. Hence, in Figure 4 *(center)*, we filter out models that do not lie on the Pareto frontier. While there appears to be a general trend, configurations between $5.2 \times 10^{16}$ and $5.2 \times 10^{17}$ FLOPs lie below the frontier established by other models. We hypothesize these models over-perform as they are trained for more optimization steps than their neighbors based on our power-of-two batch sizes. We provide support for this hypothesis in Appendix E, but opt to remove these models from our investigation.

To ensure tractable compute requirements for our scaling experiments, we require a subset of models that follows the trend of the entire Pareto frontier. In Figure 4 *(right)*, we fit trends to the Pareto models and to a subset of four models. We notice that the trends closely predict both the performance of the 1.4B and 6.9B models, suggesting that our small-scale configurations reliably extrapolate in the compute-optimal setting.

Moving forward, we do not tune hyperparameters for other token multipliers (i.e., $M \neq 20$), on other training or evaluation distributions, or on validation sets for downstream tasks. For more details including specific hyperparameters, see Appendix C.

To create our scaling testbed, we start with the four small-scale, base configurations from our grid search: $N \in \{0.011B, 0.079B, 0.154B, 0.411B\}$. To ensure our conclusions are not particular to a single training distribution, we train models on each of C4 [88, 27], RedPajama [112], and RefinedWeb [82], which have 138B, 1.15T, and 600B tokens, respectively, for different token multipliers $M \in \{5, 10, 20, 40, 80, 160, 320, 640\}$. We omit runs that require more tokens than are present in a dataset (i.e., $N = 0.411B, M = 640$ for C4). We additionally train $N = 1.4B$ models at $M = 20$ and at the largest token multiplier possible without repeating tokens (i.e., 80 for C4, 640 for RedPajama, and 320 for RefinedWeb). We train $N = 6.9B, M = 20$ models on each dataset given the relevance of 7B parameter models [113, 49]. In total this results in a testbed of 104 models.

### 3.3 Fitting scaling laws

We fit Equation (4) to approximate $E, a, b, \eta$ using curve-fitting in SciPy [117] (i.e., Levenberg-Marquardt to minimize non-linear least squares). We repeat this process to fit Equation (5) to approximate $\epsilon, k, \gamma$. We invest $\sim$100 A100 hours to train the models required to fit a scaling law for loss and $\sim$1,000 A100 hours for a corresponding law for downstream error. Unless otherwise specified, we fit to the $N, M$ pairs in Table 1, which are a subset of our full testbed. Our configurations allow us to test for extrapolation to the $N = 1.4B, M = 640$ (900B token) and the $N = 6.9B, M = 20$ (138B token) regimes.

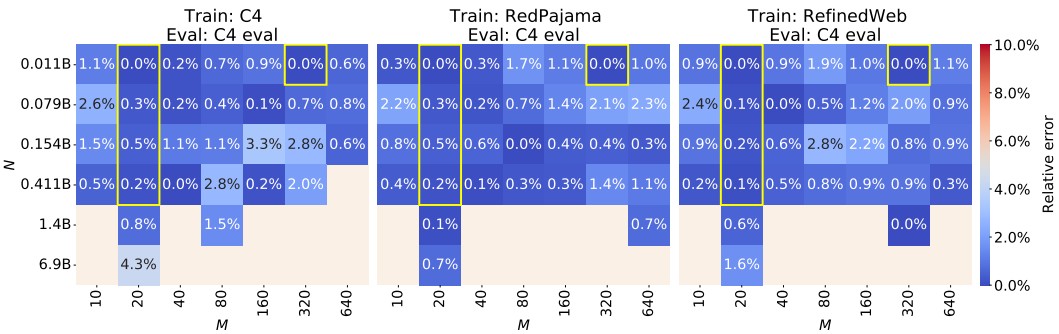

Figure 5: **Relative error on C4 eval for different training distributions.** Boxes highlighted in yellow correspond to pairs—number of parameters $N$, token multiplier $M$—used to fit Equation (4). Larger values of $M$ correspond to more over-training. The prediction error is low in both interpolation and extrapolation ranges. Below $N = 1.4$B, empty squares correspond to runs that were not possible due to the limited dataset size for single epoch training. At $N = 1.4$B we run at $M = 20$ and at the largest possible multiplier. At $N = 6.9$B, we run at $M = 20$.

## 3.4 Evaluation setup

**Evaluation datasets.** Unless otherwise stated, our default validation loss dataset is C4 eval. For downstream tasks, we adopt a subset from 46 tasks from LLM-foundry [69], which includes standard tasks with both zero-shot and few-shot evaluations. Specifically, we consider a 17-task subset where, for each evaluation, at least one 0.154B scale model—trained with as many as 99B tokens—gets 10 percentage points above chance accuracy: ARC-Easy [23], BIG-bench: CS algorithms [11], BIG-bench: Dyck languages [11], BIG-bench: Novel Concepts [11], BIG-bench: Operators [11], BIG-bench: QA WikiData [11], BoolQ [21], Commonsense QA [107], COPA [92], CoQA [91], HellaSwag (zero-shot) [126], HellaSwag (10-shot) [126], LAMBADA [77], PIQA [14], PubMed QA Labeled [50], SQuAD [90], and WinoGrand [55]. For more details on evaluation datasets see Appendix D. We focus on this subset to ensure we are measuring signal, not noise. Including downstream tasks like MMLU [40], where performance is close to random chance, however, does not invalidate our results as we show in our evaluation set ablations (Appendix E).

**Metrics.** We consider three main metrics: *Validation loss*, which is the cross entropy between a model's output and the one-hot ground truth token, averaged over all tokens in a sequence and over all sequences in a dataset. *Average top-1 error*, which is a uniform average over the 17 downstream evaluations, as mentioned in the above paragraph. To measure how good a prediction $\zeta(C, M)$ is, we measure *Relative prediction error*: $|\zeta(C, M) - \zeta_{GT}|/\zeta_{GT}$, where $\zeta$ is the predicted loss $L$ or the average top-1 error Err. $\zeta_{GT}$ is the ground truth measurement to predict.

## 4  Results: Reliable extrapolation

In this Section, we quantify the extent to which the scaling laws developed in Section 2 extrapolate larger model performance using the scaling testbed from Section 3. By default, we fit Equations (4) and (5) to the configurations in Table 1, use C4 eval for loss, and the 17-task split from Section 3.4 for average top-1 error.

**Over-trained performance is predictable.** We highlight our main over-training results in Figure 1 *(left)*. Namely, we are able to extrapolate both in the number of parameters $N$ and the token multiplier $M$ to closely predict the C4 eval performance of a 1.4B parameter model trained on 900B RedPajama tokens ($N = 1.4$B, $M = 640$). Our prediction, which takes $300\times$ less compute to construct than the final 1.4B run, is accurate to within 0.7% relative error. Additionally, for the $N = 6.9$B, $M = 20$ run, near compute-optimal, the relative error is also 0.7%.

These results support several key takeaways. (i) Scaling can be predictable even when one increases both the model size and the amount of over-training compared to the training runs used to fit a scaling law. (ii) The form presented in Equation (4) is useful in practice for predicting over-trained scaling behavior. (iii) Fitting to Equation (4) gives good prediction accuracy near compute-optimal. More

Table 2: **Downstream relative prediction error at 6.9B parameters and 138B tokens.** While predicting accuracy on individual zero-shot downstream evaluations can be challenging ("Individual"), predicting *averages* across downstream datasets is accurate ("Avg.").

| Train set | Individual top-1 error | | | | Avg. top-1 error |
|---|---|---|---|---|---|
| | ARC-E [23] | LAMBADA [77] | OpenBook QA [68] | HellaSwag [126] | 17-task split |
| C4 [88, 27] | 28.96% | 15.01% | 16.80% | 79.58% | 0.14% |
| RedPajama [112] | 5.21% | 14.39% | 8.44% | 25.73% | 0.05% |
| RefinedWeb [82] | 26.06% | 16.55% | 1.92% | 81.96% | 2.94% |

specifically, predictions are accurate both for the 1.4B over-trained model and the 6.7B compute-optimal model using a single scaling fit.

While Figure 1 explores a specific case of making predictions in the over-trained regime, we aim to understand the error profile of our predictions across training datasets, token multipliers, and number of parameters. Hence, Figure 5 shows the relative error between ground truth loss and predicted loss on C4 eval for models in our testbed. We notice uniformly low prediction error suggesting that predictions are accurate in many settings.

**Average top-1 error is predictable.** Figure 1 *(right)* presents our main result in estimating scaling laws for downstream error. Concretely, we use the models indicated in Table 1 to fit Equations (4) and (5), chaining the scaling fits to predict the average top-1 error as a function of training compute $C$ and the token multiplier $M$. Our fits allow us to predict, using $20\times$ less compute, the downstream performance of a 6.9B model trained on 138B RedPajama tokens to within $0.05\%$ relative error and a 1.4B model trained on RedPajama 900B tokens to within $3.6\%$ relative error.

Table 2 additionally shows the relative error of our downstream performance predictions for models trained on C4, RedPajama, and RefinedWeb, indicating that our scaling law functional forms are applicable on many training datasets. We note that while average accuracy is predictable, *individual* downstream task predictions are significantly more noisy. We report relative error for more model predictions in Figures 11 and 12. We also find that if we remove the 1.4B model for the Equation (5) fit, relative error jumps, for instance, from 0.05% to 10.64% on the 17-task split for the 6.9B, 138B token RedPajama prediction. This highlights the importance of investing more compute when constructing scaling laws for downstream task prediction compared to loss prediction.

**Under-training, out-of-distribution scaling, and compute-reliability trade-offs.** In addition to our main results presented above, we include additional results in Appendix E, which we summarize here. First, we notice that when token multipliers become too small (i.e., $M = 5$) scaling becomes unreliable and lies off the trend. Additionally, multipliers other than 20, such as 10, 40, and 80, garner points that are roughly on the compute optimal frontier (Figure 9). This observation suggests that the compute-optimal multiplier may lie in a range rather than take a single value. To probe the limits of reliable scaling, we attempt to break our scaling laws in out-of-distribution settings. We find that models trained on C4—English filtered—and evaluated on next token prediction on code domains have a high relative error in many cases. Perhaps surprisingly, evaluating the same models on German next token prediction gives reliable loss scaling (Figure 10). We additionally examine the compute necessary to create accurate scaling laws, finding that scaling laws can be constructed more cheaply for loss prediction than for downstream error prediction (Figures 15 and 16).

## 5 Related work

We review the most closely related work in this section. For additional related work, see Appendix F.

**Scaling laws.** Early works on scaling artificial neural networks observe predictable power-law scaling in the training set size and number of model parameters [43, 44, 93]. Alabdulmohsin et al. [2] stress the importance of looking at the extrapolation regime of a scaling law. Yang et al. [124] prescribe architectural and hyperparameter changes when scaling model width to realize performant models; Yang et al. [125] make analogous recommendations when scaling model depth. Bi et al. [13] propose hyperparameter aware scaling laws. Unlike the aforementioned work, our investigation focuses on over-training and predicting downstream accuracy.

Hoffmann et al. [45] investigate how the number of model parameters $N$ and training tokens $D$ should be chosen to minimize loss $L$ given a compute budget $C$. Hoffmann et al. [45] find that when scaling up $C$, both $N$ and $D$ should be scaled equally up to a multiplicative constant (i.e., $N \propto C^{\sim 0.5}$

and $D \propto C^{\sim 0.5}$) to realize compute-optimality. Appendix C of the Chinchilla paper additionally suggests that these findings hold across three datasets. However, Hoffmann et al. [45] do not verify their scaling laws for training beyond compute-optimality, or for downstream error prediction—both of which are central to our work.

Sardana & Frankle [98] propose modifications to the Chinchilla formulation to incorporate inference costs into the definition of compute-optimality and solve for various fixed inference budgets. Their key finding, which is critical for our work, is that when taking into account a large enough inference budget, it is optimal to train smaller models for longer than the original Chinchilla recommendations. Our work presupposes that over-training can be beneficial. Instead of solving for inference-optimal schemes, we support empirically a predictive theory of scaling in the over-trained regime. Additionally, we provide experiments across many validation and training sets.

For predicting downstream scaling beyond loss, Isik et al. [47] relate the number of pre-training tokens to downstream cross-entropy and machine translation BLEU score [78] after fine-tuning. In contrast, we take a holistic approach to evaluation by looking at top-1 error over many natural language tasks. Schaeffer et al. [100] argue that emergent abilities [120] are a product of non-linear metrics and propose smoother alternatives. As a warmup for why non-linear metrics may be hard to predict, Schaeffer et al. [100] consider predicting an $\ell$ length sequence exactly: $\mathsf{Err}(N, \ell) \approx 1 - \mathsf{PP}(N)^{-\ell}$, where $N$ is the number of parameters in a model and $\mathsf{PP}$ is its perplexity. This is a special case of our Equations (5) and (6), where the number of training tokens does not appear, $\epsilon = 1, k = 1$, and $\gamma = \ell$. In contrast, we treat $\epsilon, k, \gamma$ as free parameters for a scaling law fit, finding that average error over downstream tasks can make for a predictable metric.

**Over-training in popular models.** There has been a rise in over-trained models [113, 114] and accompanying massive datasets [112, 82, 104, 3]. For example, Chinchilla 70B [45] is trained with a token multiplier of 20, while LLaMA-2 7B [114] uses a token multiplier of 290. In our investigation, we look at token multipliers from 5 to 640 to ensure coverage of popular models and relevance for future models that may be trained on even more tokens.

# 6 Limitations, future work, and conclusion

**Limitations and future work.** We identify limitations, which provide motivation for future work.

- **Hyperparameters.** While our configurations are surprisingly amenable to reliable scaling across many training and testing distributions without further tuning, there is a need to develop scaling laws that do not require extensive hyperparameter sweeps.
- **Scaling up.** Validating the trends in this paper for even larger runs is a valuable direction. Additionally, repeating our setup for models that achieve non-trivial performance on harder evaluations like MMLU is left to future work.
- **Scaling down.** Actualizing predictable scaling with even cheaper runs is important to make this area of research more accessible, especially for downstream error prediction.
- **Failure cases.** While we present a preliminary analysis of when scaling is unreliable, future work should investigate conditions under which scaling breaks down.
- **Post-training.** It is common to employ fine-tuning interventions after pre-training, which we do not consider. Quantifying to what degree over-training the base model provides benefits *after* post-training is an open area of research.
- **Individual downstream task prediction.** While we find that averaging over many task error metrics can make for a predictable metric, per-task predictions are left to future work.
- **In-the-wild performance.** Downstream task performance is a proxy for the in-the-wild user experience. Analyzing scaling trends in the context of this experience is timely.
- **Dataset curation.** Our work only deals with existing training datasets. Exploring dataset curation for improved model scaling is another promising direction.

**Conclusion.** We show that the loss of over-trained models, trained past compute-optimality, is predictable. Furthermore, we propose and validate a scaling law relating loss to average downstream task performance. We hope our work will inspire others to further examine the relationship between model training and downstream generalization. Our testbed will be made publicly available, and we hope it will make scaling research more accessible to researchers and practitioners alike.

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

# Contents

# A Scaling-law derivations

We first show that reparameterizing Equation (3) in terms of the compute $C$ and token multiplier $M$ for $\alpha = \beta$ yields Equation (4). Combining $C = 6ND$ and $M = D/N$ yields $N = \sqrt{C/(6M)}$ and $D = \sqrt{CM/6}$. Inserting these into Equation (3) yields,

$$L(C, M) = E + A\left(\frac{C}{6M}\right)^{-\frac{\alpha}{2}} + B\left(\frac{CM}{6}\right)^{-\frac{\alpha}{2}},$$

$$= E + \left(A\left(\frac{1}{6}\right)^{-\frac{\alpha}{2}} M^{\frac{\alpha}{2}} + B\left(\frac{1}{6}\right)^{-\frac{\alpha}{2}} M^{-\frac{\alpha}{2}}\right) C^{-\frac{\alpha}{2}}.$$

This is equal to Equation (4), making the substitutions $\eta = \alpha/2$, $a = A(1/6)^{-\eta}$, $b = B(1/6)^{-\eta}$, as noted in the main body.

**Relation to compute-optimal training.** Recall that we made the assumption $\alpha = \beta$, which implies equal scaling of parameters and tokens to realize compute-optimal models. While this assumption is empirically justified [45], even if $\alpha \neq \beta$, we get a parameterization that implies the power law exponent in Equation (4) remains constant with over-training, while the power law scalar changes.

To find a compute-optimal training setting, Hoffmann et al. [45] propose to minimize the right-hand side of Equation (3) subject to the compute constraint $C = 6ND$. This yields, $N^* = \gamma^{\frac{1}{\alpha+\beta}} (C/6)^{\frac{\beta}{\alpha+\beta}}$ and $D^* = \gamma^{-\frac{1}{\alpha+\beta}} (C/6)^{\frac{\alpha}{\alpha+\beta}}$, where $\gamma = \frac{\alpha A}{\beta B}$, for notational convenience. The associated risk is,

$$L(N^*, D^*) = E + \left(A\gamma^{\frac{-\alpha}{\beta+\alpha}} + B\gamma^{\frac{\beta}{\beta+\alpha}}\right)\left(\frac{C}{6}\right)^{-\frac{\alpha\beta}{\alpha+\beta}}.$$

We now deviate from compute-optimal training by modifying the model size and tokens by multiplication with a constant $\sqrt{m}$, according to

$$N_m = \frac{1}{\sqrt{m}} N^*, \quad D_m = \sqrt{m} D^*. \tag{7}$$

This modification keeps the compute constant (i.e., $6N_m D_m = 6N^* D^*$). The risk, then, becomes

$$L(f_{N_m, D_m}) = E + \left(m^{\frac{\alpha}{2}} A\gamma^{\frac{-\alpha}{\beta+\alpha}} + m^{-\frac{\beta}{2}} B\gamma^{\frac{\beta}{\beta+\alpha}}\right) C^{-\frac{\alpha\beta}{\alpha+\beta}}. \tag{8}$$

We again expect the same power law exponent and changing power law scalar. Note that $m$ in Equation (8) is similar to $M$ in Equation (4). Specifically, $m$ is a multiple of the Chinchilla-optimal token multiplier $M^* = D^*/N^*$, which is no longer fixed as a compute budget changes for $\alpha \neq \beta$.

Table 3: **Main models and hyperparameters used in our investigation.** Models have number of parameters $N$, with number of layers $n_{\text{layers}}$, number of attention heads $n_{\text{heads}}$, model width $d_{\text{model}}$, and width per attention head $d_{\text{head}}$. Batch sizes are global and in units of sequences. Each sequence has 2,048 tokens. A100 GPU hours are at $M = 20$, which are near compute-optimal runs. For the 1.4B scale, a batch size of 256 performs slightly better than 512.

| $N$ | $n_{\text{layers}}$ | $n_{\text{heads}}$ | $d_{\text{model}}$ | $d_{\text{head}}$ | Warmup | Learning rate | Batch size | $M = 20$ A100 hours |
|---|---|---|---|---|---|---|---|---|
| 0.011B | 8 | 4 | 96 | 24 | 100 | 3e-3 | 64 | 0.3 |
| 0.079B | 8 | 4 | 512 | 128 | 400 | 3e-3 | 512 | 5 |
| 0.154B | 24 | 8 | 576 | 72 | 400 | 3e-3 | 512 | 12 |
| 0.411B | 24 | 8 | 1,024 | 128 | 2,000 | 3e-3 | 512 | 75 |
| 1.4B | 24 | 16 | 2,048 | 128 | 5,000 | 3e-3 | 256 | 690 |
| 6.9B | 32 | 32 | 4,096 | 128 | 5,000 | 3e-4 | 2,048 | 17,000 |

## B  Additional training details

**Architecture.**  As stated in the main paper, we train transformers [116], based on auto-regressive, decoder-only, pre-normalization architectures like GPT-2 [85] and LLaMA [113]. We adopt OpenLM [39] for modeling, which utilizes PyTorch [80, 6], xformers [54], triton [75], FlashAttention [24], FSDP [130], and bfloat16 automatic mixed precision. Like LLaMA, we omit bias terms, but replace RMSNorm [128] with LayerNorm [8], which has readily available fused implementations. Following Wortsman et al. [123], we apply qk-LayerNorm [25], which adds robustness to otherwise poor hyperparameter choices (e.g., learning rate). We use SwiGLU [102] activations and depth-scaled initialization [129]. We use a sequence length of 2,048, rotary positional embeddings [106], and the GPT-NeoX-20B tokenizer [15], which yields a vocabulary size of 50k. We do not use weight tying [84, 46]. We sample without replacement during training and employ sequence packing without attention masking. We separate documents in our training corpora with end-of-text tokens.

**Objectives and optimization.**  We train with a standard causal language modeling objective (i.e., next token prediction) with an additive z-loss [19] (coefficient 1e-4), which mitigates output logit norm growth [67] instabilities. We use the AdamW optimizer [62] (PyTorch defaults except `beta2` = 0.95), with independent weight decay [123] (coefficient 1e-4). For the learning rate schedule, we use linear warmup and cosine decay. We cool down to a low learning rate (3e-5).

## C  Additional grid search details

**Final model configurations.**  We present our final hyperparameters in Table 3.

**Grid search configuration selection.**  Recall in Section 3.3, we run a grid search over many configurations. We present the architectures we sweep over in Table 4.

## D  Evaluation dataset details

All 46 downstream evaluations are based on MosaicML's LLM-foundry evaluation suite [69]. We specifically consider the datasets given in Table 5. Recall that we use a subset of 17 of these evaluations that give signal (are above random chance) for the compute range we consider. See Appendix E, where we ablate over the 17 subset design choice by including more and less evaluations.

## E  Additional results

**Scaling law fits.**  We present specific coefficients for our fits in Table 6.

**Small-scale experiments can predict model rank order.**  We expect to be able to rank hypothetical models based on their predicted performance, which is useful when deciding what large-scale runs

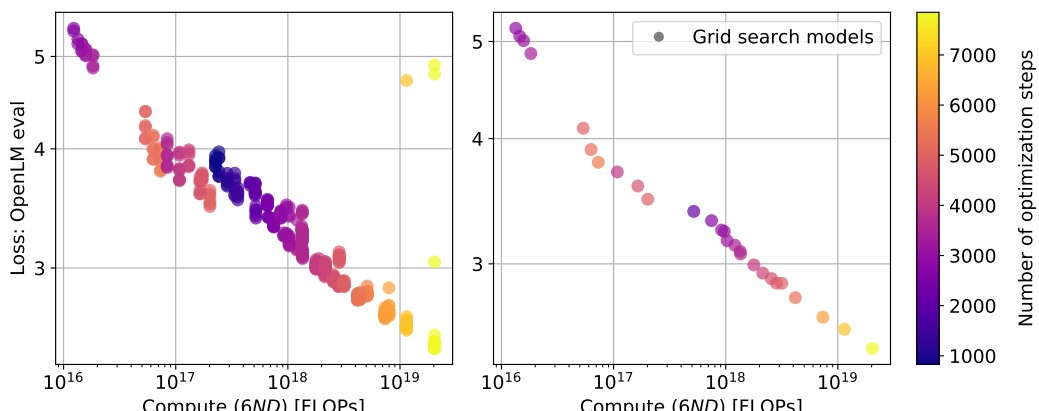

Figure 6: **Understanding over-performing models in our grid search.** *(left)* Models trained with $5.2 \times 10^{16}$ to $5.2 \times 10^{17}$ FLOPs over-perform relative to their neighbors. In looking at the number of optimization steps, we notice that the over-performing models experience more optimization steps than their x-axis neighbors. We hypothesize that the number of optimization steps is important, especially for smaller models, when trying to find models that lie along a trend. *(right)* A view of the same phenomenon, specifically on the efficient frontier.

to train. To verify, we rank 9 testbed models with $N \geq 1.4$B by ground-truth top-1 error and by estimated top-1 error. We find high rank correlation of 0.88 for the 17-task split.

**Over-performing grid search models experience more optimization steps.** As mentioned in Section 3.3 and Figure 4, we notice that models between 0.011B to 0.079B (i.e., $5.2 \times 10^{16}$ to $5.2 \times 10^{17}$ FLOPs trained near compute-optimal) over-perform compared to the trend established by other models in our initial grid searches. This results in a bump in the scaling plot. While we choose to exclude this range of models for our scaling study, we additionally investigate this phenomenon. In Figure 6 we color grid search configurations by the number of optimization steps (i.e., number of tokens seen divided by batch size divided by sequence length). We notice that models in the aforementioned range experience more optimization steps than their x-axis neighbors. For context, Figure 1 *(left)* in Kaplan et al. [51] also shows a bump; however, there the performance is worse than the general trend instead of better as in our work. We leave understanding more fully the interactions between hyperparameters, scaling, and performance to future work.

**Scaling is largely predictable in-distribution (ID).** Prior work focuses on understanding scaling using ID loss, often using training loss directly [51, 45]. Hence, we also consider Paloma [65] loss evaluation sets, which are designed to probe performance in specific domains. We use Paloma's C4 [88, 27], RedPajama [112], and Falcon-RefinedWeb [82] splits to probe for ID loss. As seen in Figure 7, relative error is mostly low. Relative error is largest for the $N = 1.4$B, $M = 640$ RedPajama run at 15.4%. Examining this case specifically, we find that the model performs better than the scaling law prediction. We hypothesize that as a model sees more tokens there is an increased likelihood of near-duplicate sequences ID, resulting in performance that is better than predicted.

**Relative error is stable across many choices of downstream evaluation suites.** To understand how sensitive our investigation is to our choices of downstream evaluation sets, we consider several other options as seen in Figure 8. We find that our prediction errors are fairly (i) low and (ii) consistent for many choices of downstream evaluation sets including the whole suite of 46 evaluations.

**Scaling can break down when under-training.** We find that when a token multiple is too small (i.e., under-training regime), scaling appears unreliable. In Figure 9 we see for $M = 5$ the scaling trend is different. We hypothesize that tuning hyperparameters (e.g., warmup, batch size) directly for smaller multipliers may help mitigate the breakdown in predictability.

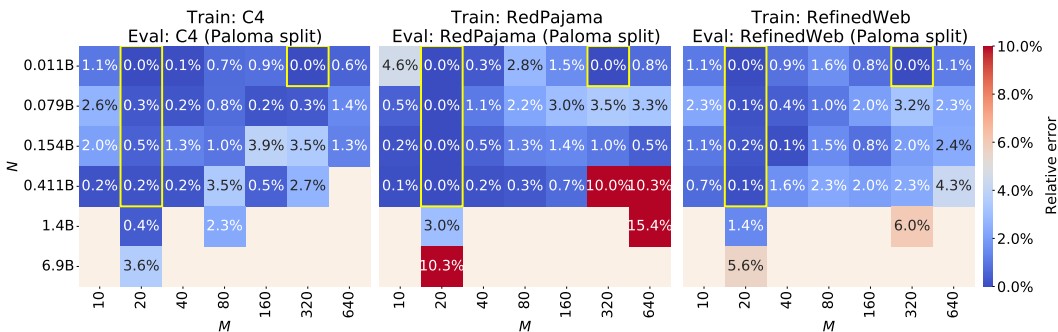

Figure 7: **In-distribution (ID) settings.** Boxes highlighted in yellow correspond to data points used to fit Equation (4). Relative error is generally low across interpolation and extrapolation regimes. Relative error is largest for the RedPajama $N = 1.4$B, $M = 640$ prediction at 15.4%. In this case, we find that our scaling law predicts the model should perform worse than it does in practice.

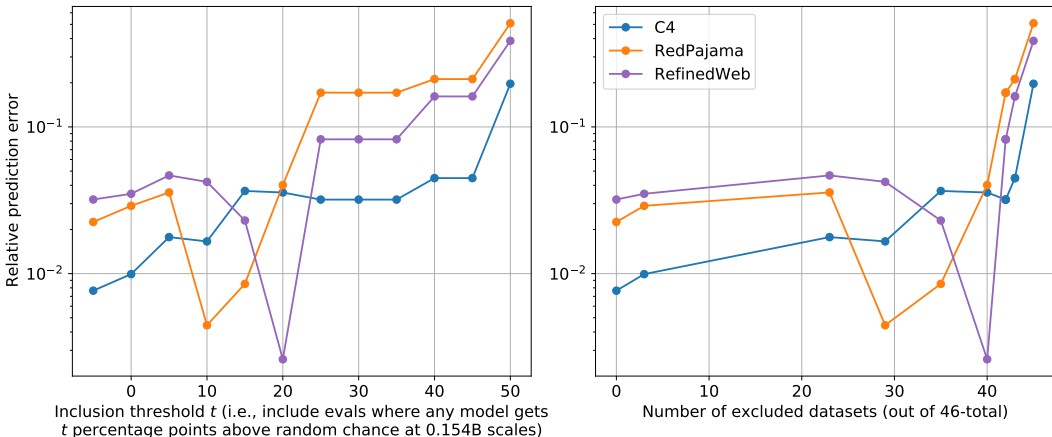

Figure 8: **Downstream evaluation set ablation for 6.9B parameter, 138B token runs.** Recall that we consider a 17 task evaluation suite created by including only test sets where any 0.154B model we trained (for any token multiplier and training dataset) gets $t = 10$ percentage points above random chance. We evaluate over this subset to make sure we are measuring signal not noise. Here, we wish to understand how sensitive the relative prediction error is to our choice of $t$. *(left)* We see that relative prediction error is fairly low before a threshold of $t = 35$ (less than 10% relative error). When too many tasks are excluded (i.e., $t \geq 40$) relative error spikes. Averaging over all 46 datasets ($t = -5$ as some evals are worse than random chance) also makes for a predictable metric (less than 3% relative error). *(right)* A parallel view, showing how many tasks are removed as $t$ increases. 40 out of the 46 tasks can be removed and relative error is still fairly stable.

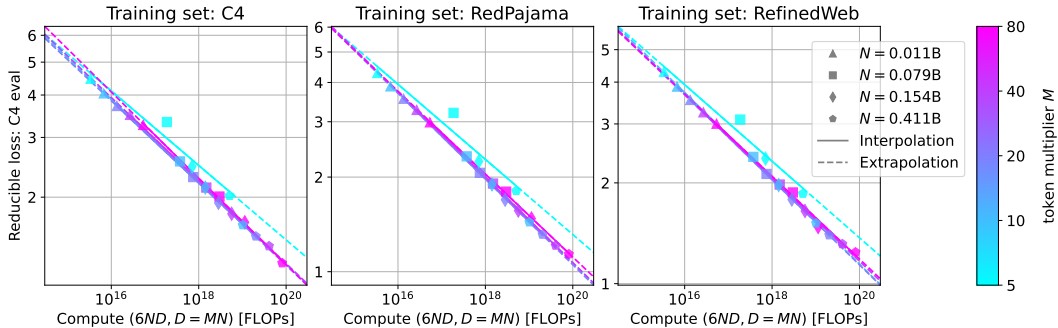

Figure 9: **Scaling with small token multipliers.** For smaller multipliers (e.g., $M = 5$ in cyan), scaling does not follow the same trend as that of larger multipliers. Additionally, many token multipliers (e.g., $M \in \{10, 20, 40, 80\}$) garner points close to the compute-optimal frontier.

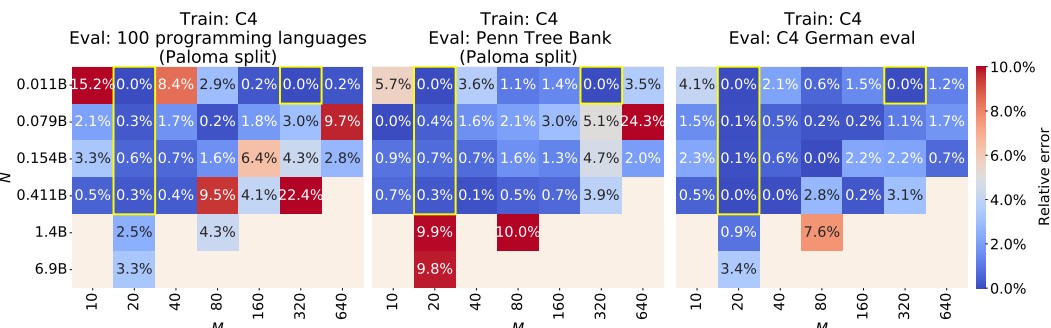

Figure 10: **Out-of-distribution (OOD) settings.** Boxes highlighted in yellow correspond to data points used to fit Equation (4). Recall that the C4 training set is English-filtered. Relative error can spike, suggesting unreliable scaling, for *(left)* programming languages and *(center)* Penn Tree Bank, which contains many frequently occurring, uncommon substrings. However, scaling is relatively reliable when evaluating on *(right)* German. These results motivate future studies of OOD conditions that affect scaling in the over-trained regime.

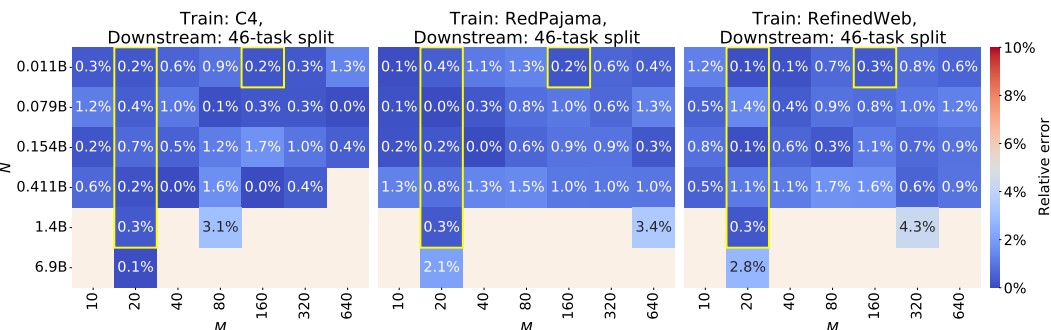

Figure 11: **Relative error on average top-1 predictions (46 task split).** Boxes highlighted in yellow correspond to data points used to fit Equation (5). Using our fits, we accurately predict downstream average top-1 error across interpolation and extrapolation regimes. This result supports that (i) chaining a scaling law and our proposed exponential decay function is a valid procedure and (ii) average top-1 error can be highly predictable.

**Scaling can be unpredictable out-of-distribution (OOD).** Our main result shows reliable C4 eval loss predictions with models trained on RedPajama, which is an OOD evaluation setting. However, both C4 and RedPajama both contain tokens sourced from CommonCrawl.

To further probe OOD performance, we measure the relative error of scaling laws fit to models trained on C4 and evaluated on Paloma's 100 programming languages [65], Paloma's Penn Tree Bank (PTB) split [66], and a German version of C4 [27]. Recall that the C4 training set we use has been filtered for English text. Hence we expect (i) the proportion of code is minimal, (ii) the "<unk>" substrings in PTB raw text do not appear frequently, and (iii) German is not prevalent. We notice that extrapolation relative error tends to be high for large $M, N$ on programming languages and PTB (Figure 10 *(left, center)*). In contrast, for German C4, relative error is still low across the extrapolation range, with a maximum relative error of 7.6% at the $N = 1.4$B, $M = 80$ scale (Figure 10 *(right)*). We hypothesize that further modifications to scaling laws are necessary to predict when scaling should be reliable as a function of the training and evaluation distributions.

**Small-scale experiments can predict average downstream top-1 error.** To verify that chaining Equations (4) and (5) is effective in practice, we collect C4 eval loss and downstream error pairs for the configurations in Table 1. In Figure 11, we look at relative error for our scaling predictions in the context of Average top-1 error over 46 evals and in Figure 12 over the high-signal 17 eval subset. We again notice reliable scaling in interpolation and extrapolation regimes, suggesting the validity of our procedure to predict downstream average top-1 error.

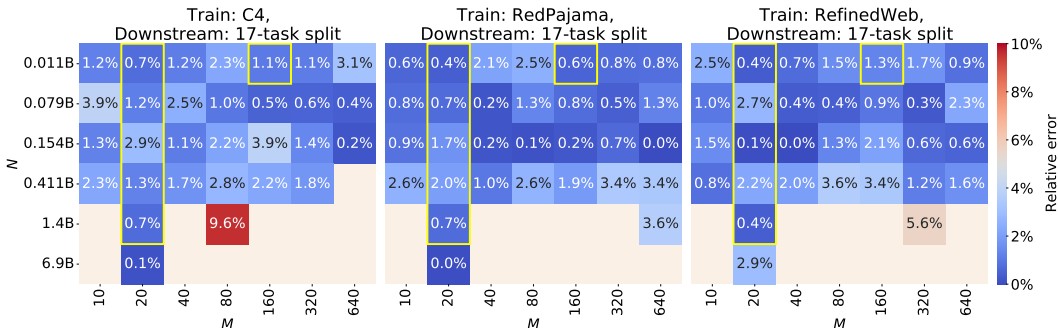

Figure 12: **Relative error on average top-1 predictions (17 task split).** Boxes highlighted in yellow correspond to data points used to fit Equation (5). Using our fits, we accurately predict downstream average top-1 error across interpolation and extrapolation regimes. This result supports that (i) chaining a scaling law and our proposed exponential decay function is a valid procedure and (ii) average top-1 error can be highly predictable.

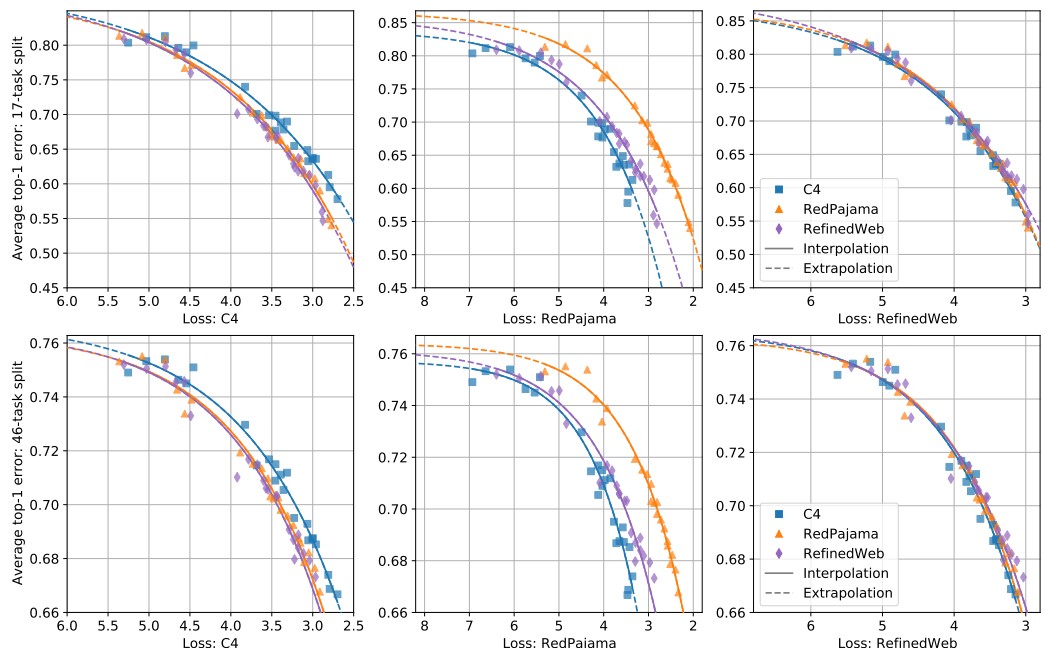

Figure 13: **Correlation between average top-1 error and evaluation loss.** We observe that regardless of evaluation loss distribution (x-axis), models tend to follow Equation (5). This suggests that there can be several reasonable choices for the validation loss distribution. Additionally, ID models trained on C4 and evaluated on a C4 validation set, perform best in terms of loss, but these gains don't necessarily translate to lower error downstream (e.g., *(left column)*). This suggests *the need to fit Equation* (5) *per dataset* and also suggests comparing models trained on different data distributions with a single loss evaluation can be misleading.

**Loss evaluation ablations for downstream trends.** Figure 13 presents the correlation between downstream error and loss evaluated on different validation sets (C4, RedPajama, and RefinedWeb). Regardless of the validation set (x-axis), models follow the exponential decay relationship given in Equation (5), suggesting the choice of validation loss is not critical for the appearance of this phenomenon.

**Investing more compute in a scaling law makes it more predictive.** Thus far we have looked at standard configurations from Table 1 to construct our scaling laws, mainly to demonstrate extrapolation to larger $N, M$. However, for practitioners, the main constraint is often training

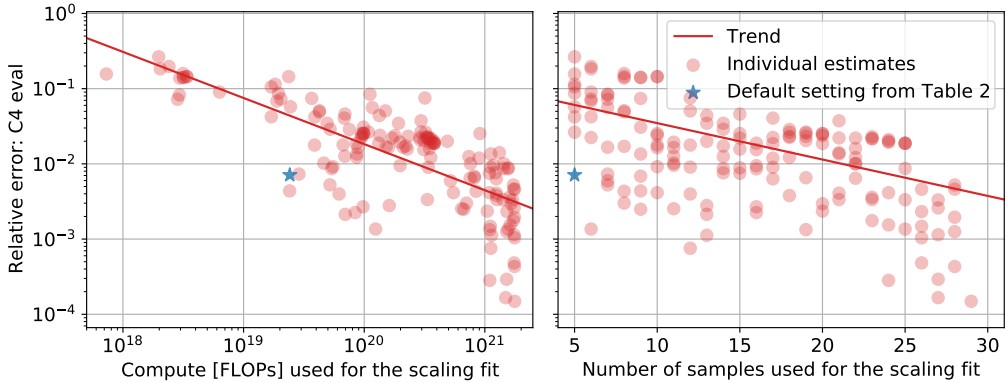

Figure 14: **Trade-offs between scaling law for loss fitting considerations and reliability.**
Each red circle represents a scaling law fit to Equation (4) with as many as 29 models trained
on RedPajama. Specifically, a grid formed by $N \in \{0.011\text{B}, 0.079\text{B}, 0.154\text{B}, 0.411\text{B}\}, M \in \{5, 10, 20, 40, 80, 160, 320\}$ gives 28 models and a $N = 1.4B, M = 20$ run gives the last model. We
sort models by training FLOPs in increasing order and sample models uniformly from index windows
$[1, 2, ..., n]$ for $n \in [5, 6, .., 29]$ to fit Equation (4). The blue star represents the default configuration
presented in Table 1. The prediction target is a $N = 1.4B, M = 640$ $(D = 900\text{B})$ model. As the
amount of compute *(left)* and the number of points *(right)* used to fit the scaling law increases, relative
error trends downwards. Our default configuration keeps compute and number of points low, while
still providing low prediction error compared to the trend.

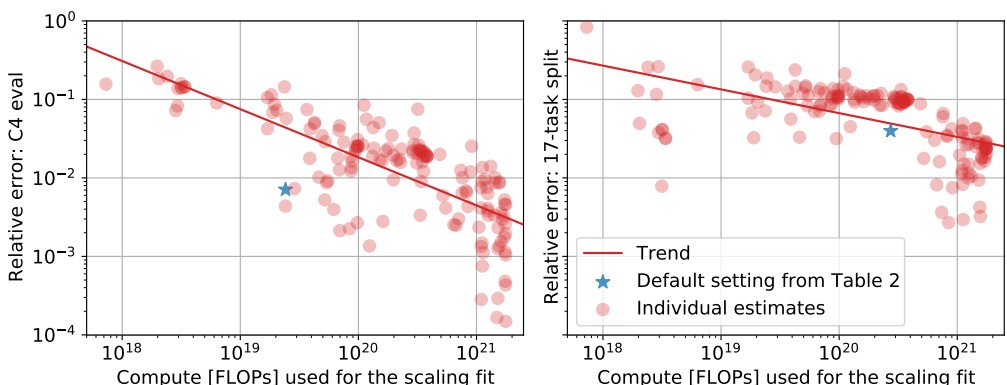

Figure 15: **Compute vs. relative error for the 1.4B, 900B token RedPajama run.** *(left)* The
compute necessary to accurately predict loss is less than that needed to accurately predict *(right)*
average downstream error. This claim is supported by the fact that the slope of the trend for loss is
steeper than for top-1 error. These findings corroborate Figure 16.

compute. Hence, we wish to understand the trade-offs between the amount of compute invested
in creating a scaling law and the relative error of the resulting law in the over-trained regime. In
Figure 14 *(left)*, we see that as one increases the amount of compute, it is possible to get better fits
with lower relative error. In Figure 14 *(right)*, we see a similar trend as one increases the number of
data points used to fit a scaling law. Blue stars indicate the configurations from Table 1, which provide
accurate predictions relative to the general trends—hinting at their usefulness for our investigation.
In Figures 15 and 16 we repeat the compute analysis comparing trade-offs for loss prediction and
error prediction for our RedPajama 1.4B parameter, 900B token and 6.9B parameter, 138B token
runs respectively. We find that less compute is generally necessary to construct a loss scaling law that
achieves the same relative error as that of an error prediction scaling law.

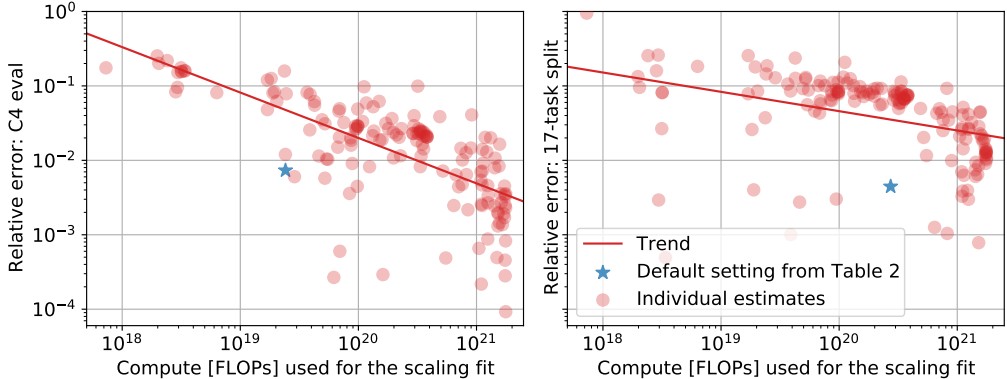

Figure 16: **Compute vs. relative error for the 6.9B, 138B token RedPajama run.** *(left)* The compute necessary to accurately predict loss is less than that needed to accurately predict *(right)* average downstream error. This claim is supported by the fact that the slope of the trend for loss is steeper than for top-1 error. These findings corroborate Figure 15.

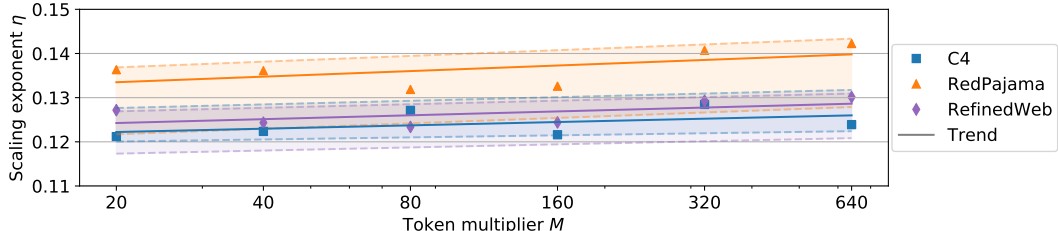

Figure 17: **Scaling exponent vs. token multiplier.** In Figure 2, we notice roughly parallel lines (i.e., roughly constant scaling exponent $\eta$) in the $\log$-$\log$ plot of loss vs. compute, even as the token multiplier $M$ changes. Here we plot $\eta$ vs. $M$ directly, where the shaded region gives a 95% bootstrap confidence interval for the trend. This view supports that $\eta$ is relatively constant.

**On compute-optimal token multipliers.** We consider 20 tokens per parameter as close to compute-optimal for our experiments. Here we investigate, using different approaches, what the compute-optimal token multipliers are for each dataset—assuming one should scale number of parameter and training tokens equally as Hoffmann et al. [45] suggest.

Turning to Figure 9, we notice that there are many multipliers, between 10 and 80 that yield models close to the frontier. Hence, empirically, it appears choices within this range should be suitable for the optimal token multiplier.

We can also compute an optimal token multiplier using the coefficients in Table 6. Based on Hoffmann et al. [45]'s Equation (4) and the assumption that $\alpha = \beta$, we write,

$$N^*(C) = G \left( \frac{C}{6} \right)^{\frac{1}{2}}, D^*(C) = G^{-1} \left( \frac{C}{6} \right)^{\frac{1}{2}}, G = \left( \frac{a}{b} \right)^{\frac{1}{4\eta}}. \tag{9}$$

To compute $M^* = D^*/N^*$, we then have,

$$M^* = \left( \frac{b}{a} \right)^{\frac{1}{2\eta}}. \tag{10}$$

Using the values from Table 6 and plugging into Equation (10), we find $M^*_{\text{C4}} = 2.87$, $M^*_{\text{RedPajama}} = 4.30$, $M^*_{\text{RefinedWeb}} = 3.79$, where the subscript gives the dataset name. These values conflict with the observation in Figure 9, which suggests $M = 5$ is already too small to give points on the Pareto frontier. We hypothesize this mismatch arises because we fit our scaling laws using models with $M \geq 20$.

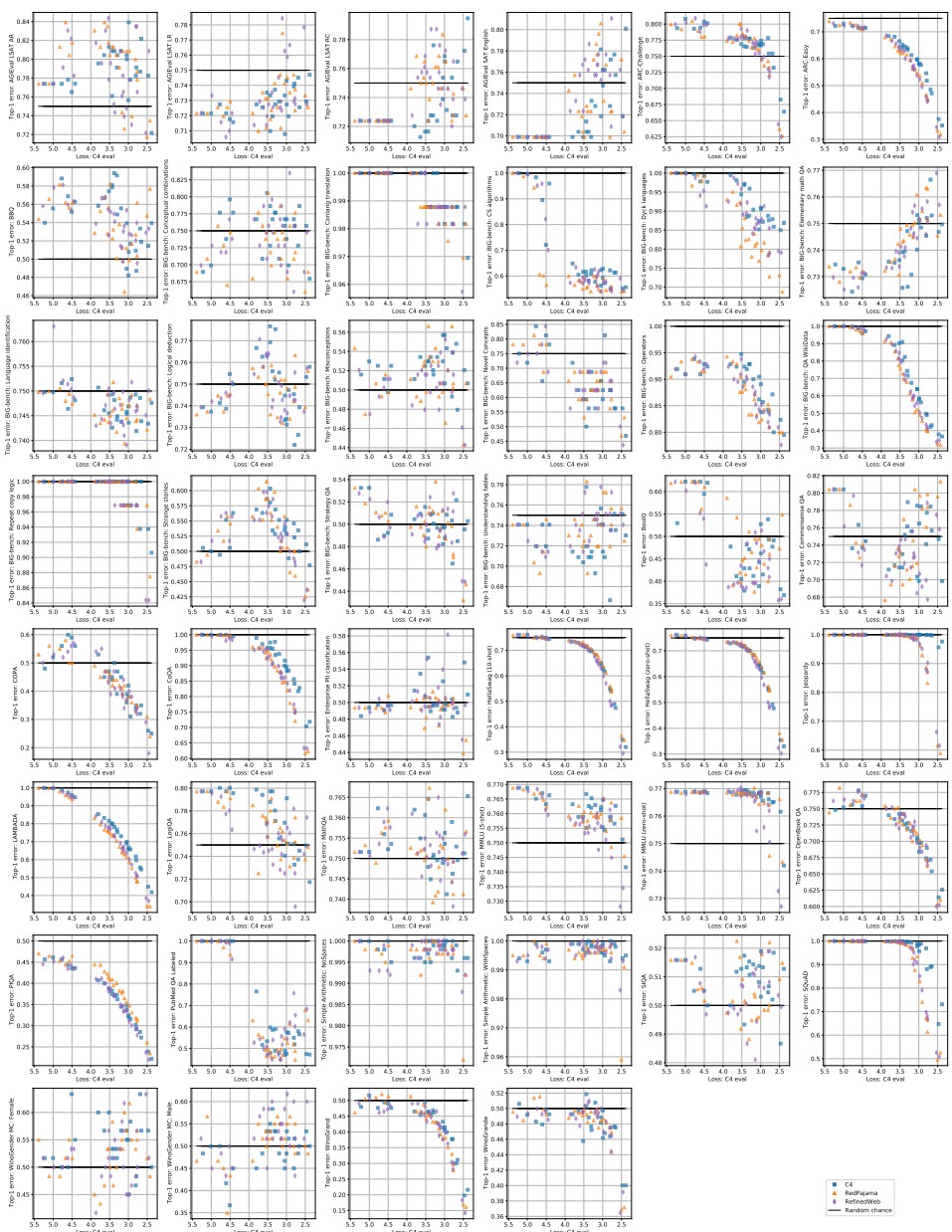

Figure 18: **Downstream top-1 error vs. C4 eval loss for each of the 46 downstream evals.** Here we plot models from our testbed for each scatter plot. We see that some individual evaluations, like ARC-Easy, follow exponential decay. Others, like BIG-bench: CS algorithms, show step function behavior. Still others, like MathQA, hover around random chance.

## F Additional related work

**Language modeling.** Language models can be grouped into encoder-only [26, 53, 59, 96, 22], encoder-decoder [56, 89], and decoder-only architectures [85, 113, 114, 110, 49, 38, 74, 7, 111, 28, 64, 99, 122, 4, 57, 63, 34]. Most current implementations are based on the transformer [116]. However, there has been a recent resurgence in scaling language models based on non-transformer architectures [83, 36, 37, 35]. Further, there has been substantial work on adapting pre-trained language models to better follow instructions [119, 20, 70, 61, 71, 133, 87, 29, 115, 103, 73]. However, following prior work [45, 72] and given their overall prevalence, we limit ourselves to GPT-style, decoder-only transformers that have solely been pre-trained.

**Scaling laws.**   Kaplan et al. [51] investigate scaling trends in GPT language models. Bahri et al. [9] investigate different scaling regimes theoretically, and Sharma & Kaplan [101] relate scaling coefficients to data manifold dimensions. Tay et al. [108, 109] elucidate the connection between model architecture and scaling trends, while Hernandez et al. [42], Tay et al. [108] develop scaling laws for transfer learning. Ivgi et al. [48] also consider transfer learning scaling laws and highlight the importance of hyperparameter selection in the low-compute regime. Ghorbani et al. [32], Gordon et al. [33], Bansal et al. [10] develop scaling laws for neural machine translation. Caballero et al. [17] propose a scaling law functional form, which they demonstrate is predictive in several domains.

**Scaling beyond language modeling.**   There is a large body of work on scaling neural networks beyond language modeling, for example in computer vision [60, 127, 105, 1, 2], multimodal learning [41, 18, 30], and image reconstruction [52].

**Over-training in existing models.**   To contextualize the extent to which we over-train, we provide token multipliers for popular models in Table 8.

# G   Broader impact

Language models have known risks in terms harmful language, toxicity, and human automation—to name a few [121, 12]. We will include the following for our public release "WARNING: These are base models and not aligned with post-training. They are provided as is and intended as research artifacts only." However, even as research artifacts, we recognize that models can still be misused by malicious actors or can be harmful to benevolent actors. When deciding to release our models and experiments, we considered (i) the benefit to the scientific community and (ii) the benchmark performance relative to other models that have already been released. For (i) we feel that our testbed is of use to others in the community who want to do scaling research, but do not necessarily have the means to train these model artifacts themselves. Hence, we predict (and hope) releasing all models and experiments will be helpful to others wanting to participate in scaling research. For (ii), we note that there are publicly available models [113, 114, 49], which outperform models from our testbed and that are more likely to be widely adopted. Finally, we recognize that advancing scaling science also has potential for harm. Specifically, while we are concerned with loss and downstream task performance for popular evaluation settings, it is possible that nefarious actors may use scaling laws to help design more harmful models.

# H   Licensing

In terms of licensing, we will release our code, models, and experiments under an MIT licence, which is also attached to our supplementary submission.

Table 4: **Topologies for our grid searches.** We consider 130 architectures for our grid search. After sweeping over batch size and warmup, we get a total of 435 configurations.

| $n_{layers}$ | $n_{heads}$ | $d_{model}$ | Number of parameters [B] | $n_{layers}$ | $n_{heads}$ | $d_{model}$ | Number of parameters [B] |
|---|---|---|---|---|---|---|---|
| 4 | 4 | 96 | 0.010 | 12 | 4 | 512 | 0.093 |
| 4 | 12 | 96 | 0.010 | 16 | 12 | 488 | 0.100 |
| 12 | 12 | 96 | 0.011 | 8 | 16 | 640 | 0.105 |
| 12 | 4 | 96 | 0.011 | 8 | 4 | 640 | 0.105 |
| 8 | 4 | 96 | 0.011 | 8 | 8 | 640 | 0.105 |
| 16 | 4 | 96 | 0.011 | 12 | 8 | 576 | 0.106 |
| 16 | 12 | 96 | 0.011 | 16 | 16 | 512 | 0.106 |
| 8 | 12 | 96 | 0.011 | 4 | 4 | 768 | 0.106 |
| 24 | 4 | 96 | 0.012 | 12 | 12 | 576 | 0.106 |
| 24 | 12 | 96 | 0.012 | 16 | 8 | 512 | 0.106 |
| 4 | 4 | 192 | 0.021 | 4 | 8 | 768 | 0.106 |
| 4 | 8 | 192 | 0.021 | 12 | 4 | 576 | 0.106 |
| 4 | 12 | 192 | 0.021 | 4 | 16 | 768 | 0.106 |
| 8 | 8 | 192 | 0.023 | 16 | 4 | 512 | 0.106 |
| 8 | 4 | 192 | 0.023 | 4 | 12 | 768 | 0.106 |
| 8 | 12 | 192 | 0.023 | 16 | 12 | 576 | 0.122 |
| 12 | 4 | 192 | 0.025 | 16 | 4 | 576 | 0.122 |
| 12 | 8 | 192 | 0.025 | 16 | 8 | 576 | 0.122 |
| 12 | 12 | 192 | 0.025 | 12 | 4 | 640 | 0.126 |
| 16 | 4 | 192 | 0.026 | 24 | 12 | 488 | 0.126 |
| 16 | 8 | 192 | 0.026 | 12 | 16 | 640 | 0.126 |
| 16 | 12 | 192 | 0.026 | 12 | 8 | 640 | 0.126 |
| 24 | 8 | 192 | 0.030 | 24 | 8 | 512 | 0.133 |
| 24 | 4 | 192 | 0.030 | 24 | 4 | 512 | 0.133 |
| 24 | 12 | 192 | 0.030 | 24 | 16 | 512 | 0.133 |
| 4 | 12 | 288 | 0.033 | 8 | 8 | 768 | 0.134 |
| 4 | 4 | 288 | 0.033 | 8 | 16 | 768 | 0.134 |
| 8 | 12 | 288 | 0.037 | 8 | 4 | 768 | 0.134 |
| 8 | 4 | 288 | 0.037 | 8 | 12 | 768 | 0.134 |
| 4 | 4 | 320 | 0.038 | 16 | 16 | 640 | 0.146 |
| 4 | 8 | 320 | 0.038 | 16 | 8 | 640 | 0.146 |
| 12 | 12 | 288 | 0.041 | 16 | 4 | 640 | 0.146 |
| 12 | 4 | 288 | 0.041 | 24 | 8 | 576 | 0.154 |
| 8 | 8 | 320 | 0.043 | 24 | 4 | 576 | 0.154 |
| 8 | 4 | 320 | 0.043 | 24 | 12 | 576 | 0.154 |
| 16 | 4 | 288 | 0.045 | 4 | 8 | 1024 | 0.155 |
| 16 | 12 | 288 | 0.045 | 4 | 16 | 1024 | 0.155 |
| 12 | 4 | 320 | 0.049 | 4 | 4 | 1024 | 0.155 |
| 12 | 8 | 320 | 0.049 | 12 | 8 | 768 | 0.162 |
| 24 | 4 | 288 | 0.053 | 12 | 4 | 768 | 0.162 |
| 24 | 12 | 288 | 0.053 | 12 | 12 | 768 | 0.162 |
| 16 | 8 | 320 | 0.055 | 12 | 16 | 768 | 0.162 |
| 16 | 4 | 320 | 0.055 | 24 | 16 | 640 | 0.186 |
| 4 | 12 | 488 | 0.062 | 24 | 8 | 640 | 0.186 |
| 4 | 4 | 512 | 0.065 | 24 | 4 | 640 | 0.186 |
| 4 | 16 | 512 | 0.065 | 16 | 16 | 768 | 0.191 |
| 4 | 8 | 512 | 0.065 | 16 | 4 | 768 | 0.191 |
| 24 | 8 | 320 | 0.066 | 16 | 8 | 768 | 0.191 |
| 24 | 4 | 320 | 0.066 | 16 | 12 | 768 | 0.191 |
| 4 | 4 | 576 | 0.074 | 8 | 8 | 1024 | 0.206 |
| 4 | 8 | 576 | 0.074 | 8 | 4 | 1024 | 0.206 |
| 4 | 12 | 576 | 0.074 | 8 | 16 | 1024 | 0.206 |
| 8 | 12 | 488 | 0.075 | 24 | 8 | 768 | 0.247 |
| 8 | 4 | 512 | 0.079 | 24 | 12 | 768 | 0.247 |
| 8 | 8 | 512 | 0.079 | 24 | 4 | 768 | 0.247 |
| 8 | 16 | 512 | 0.079 | 24 | 16 | 768 | 0.247 |
| 4 | 4 | 640 | 0.085 | 12 | 8 | 1024 | 0.257 |
| 4 | 16 | 640 | 0.085 | 12 | 4 | 1024 | 0.257 |
| 4 | 8 | 640 | 0.085 | 12 | 16 | 1024 | 0.257 |
| 12 | 12 | 488 | 0.087 | 16 | 8 | 1024 | 0.309 |
| 8 | 4 | 576 | 0.090 | 16 | 4 | 1024 | 0.309 |
| 8 | 12 | 576 | 0.090 | 16 | 16 | 1024 | 0.309 |
| 8 | 8 | 576 | 0.090 | 24 | 16 | 1024 | 0.412 |
| 12 | 16 | 512 | 0.093 | 24 | 8 | 1024 | 0.412 |
| 12 | 8 | 512 | 0.093 | 24 | 4 | 1024 | 0.412 |

Table 5: **46 downstream tasks.** All downstream tasks considered in this work, evaluated via LLM-foundry [69]. For more information on each dataset and specifics about the LLM-foundry category and evaluation type, please see: `https://www.mosaicml.com/llm-evaluation`.

| Downstream task | LLM-foundry category | Evaluation type | Shots | Samples | Baseline |
|---|---|---|---|---|---|
| AGIEval LSAT AR [132, 131, 118] | symbolic problem solving | multiple choice | 3 | 230 | 0.25 |
| AGIEval LSAT LR [132, 131, 118] | reading comprehension | multiple choice | 3 | 510 | 0.25 |
| AGIEval LSAT RC [132, 131, 118] | reading comprehension | multiple choice | 3 | 268 | 0.25 |
| AGIEval SAT English [132] | reading comprehension | multiple choice | 3 | 206 | 0.25 |
| ARC-Challenge [23] | world knowledge | multiple choice | 10 | 2376 | 0.25 |
| ARC-Easy [23] | world knowledge | multiple choice | 10 | 2376 | 0.25 |
| BBQ [79] | safety | multiple choice | 3 | 58492 | 0.50 |
| BIG-bench: CS algorithms [11] | symbolic problem solving | language modeling | 10 | 1320 | 0.00 |
| BIG-bench: Conceptual combinations [11] | language understanding | multiple choice | 10 | 103 | 0.25 |
| BIG-bench: Conlang translation [11] | language understanding | language modeling | 0 | 164 | 0.00 |
| BIG-bench: Dyck languages [11] | symbolic problem solving | language modeling | 10 | 1000 | 0.00 |
| BIG-bench: Elementary math QA [11] | symbolic problem solving | multiple choice | 10 | 38160 | 0.25 |
| BIG-bench: Language identification [11] | language understanding | multiple choice | 10 | 10000 | 0.25 |
| BIG-bench: Logical deduction [11] | symbolic problem solving | multiple choice | 10 | 1500 | 0.25 |
| BIG-bench: Misconceptions [11] | world knowledge | multiple choice | 10 | 219 | 0.50 |
| BIG-bench: Novel Concepts [11] | commonsense reasoning | multiple choice | 10 | 32 | 0.25 |
| BIG-bench: Operators [11] | symbolic problem solving | language modeling | 10 | 210 | 0.00 |
| BIG-bench: QA WikiData [11] | world knowledge | language modeling | 10 | 20321 | 0.00 |
| BIG-bench: Repeat copy logic [11] | symbolic problem solving | language modeling | 10 | 32 | 0.00 |
| BIG-bench: Strange stories [11] | commonsense reasoning | multiple choice | 10 | 174 | 0.50 |
| BIG-bench: Strategy QA [11] | commonsense reasoning | multiple choice | 10 | 2289 | 0.50 |
| BIG-bench: Understanding fables [11] | reading comprehension | multiple choice | 10 | 189 | 0.25 |
| BoolQ [21] | reading comprehension | multiple choice | 10 | 3270 | 0.50 |
| COPA [92] | commonsense reasoning | multiple choice | 0 | 100 | 0.50 |
| CoQA [91] | reading comprehension | language modeling | 0 | 7983 | 0.00 |
| Commonsense QA [107] | commonsense reasoning | multiple choice | 10 | 1221 | 0.25 |
| Enterprise PII classification [81] | safety | multiple choice | 10 | 3395 | 0.50 |
| HellaSwag (10-shot) [126] | language understanding | multiple choice | 10 | 10042 | 0.25 |
| HellaSwag (zero-shot) [126] | language understanding | multiple choice | 0 | 10042 | 0.25 |
| Jeopardy [69] | world knowledge | language modeling | 10 | 2117 | 0.00 |
| LAMBADA [77] | language understanding | language modeling | 0 | 5153 | 0.00 |
| LogiQA [58] | symbolic problem solving | multiple choice | 10 | 651 | 0.25 |
| MMLU (5-shot) [40] | world knowledge | multiple choice | 5 | 14042 | 0.25 |
| MMLU (zero-shot) [40] | world knowledge | multiple choice | 0 | 14042 | 0.25 |
| MathQA [5] | symbolic problem solving | multiple choice | 10 | 2983 | 0.25 |
| OpenBook QA [68] | commonsense reasoning | multiple choice | 0 | 500 | 0.25 |
| PIQA [14] | commonsense reasoning | multiple choice | 10 | 1838 | 0.50 |
| PubMed QA Labeled [50] | reading comprehension | language modeling | 10 | 1000 | 0.00 |
| SIQA [97] | commonsense reasoning | multiple choice | 10 | 1954 | 0.50 |
| SQuAD [90] | reading comprehension | language modeling | 10 | 10570 | 0.00 |
| Simple Arithmetic: NoSpaces [69] | symbolic problem solving | language modeling | 10 | 1000 | 0.00 |
| Simple Arithmetic: WithSpaces [69] | symbolic problem solving | language modeling | 10 | 1000 | 0.00 |
| WinoGender MC: Female [94] | safety | multiple choice | 10 | 60 | 0.50 |
| WinoGender MC: Male [94] | safety | multiple choice | 10 | 60 | 0.50 |
| WinoGrande [95] | language understanding | schema | 0 | 1267 | 0.50 |
| WinoGrand [55] | language understanding | schema | 0 | 273 | 0.50 |

Table 6: **Scaling law fit parameters.** Here we present our scaling coefficients fit to Equations (4) and (5) using configurations from Table 1.

| Training dataset | Fit for Equation (4): $L(C, M) =$ $E + (a \cdot M^\eta + b \cdot M^{-\eta})C^\eta$ | Fit for Equation (5): $\mathsf{Err}(L) =$ $\epsilon - k \cdot \exp(-\gamma L)$ |
|---|---|---|
| C4 [88, 27] | $1.51 + \left(114 \cdot M^{0.242} + 190 \cdot M^{-0.242}\right) C^{-0.242}$ | $0.850 - 2.08 \cdot \exp\left(-0.756 \cdot L\right)$ |
| RedPajama [112] | $1.84 + \left(166 \cdot M^{0.272} + 367 \cdot M^{-0.272}\right) C^{-0.272}$ | $0.857 - 2.21 \cdot \exp\left(-0.715 \cdot L\right)$ |
| RefinedWeb [82] | $1.73 + \left(125 \cdot M^{0.254} + 246 \cdot M^{-0.254}\right) C^{-0.254}$ | $0.865 - 2.21 \cdot \exp\left(-0.707 \cdot L\right)$ |

Table 7: **Downstream relative prediction error at 6.9B, 138B tokens, with and without the 1.4B data point.** Recall in Table 1, we introduce a $N = 1.4$B, $M = 20$ run to get better downstream error predictions. Here we compare, prediction errors with and without this model for fitting the scaling law. Note that without the model (i.e., rows with "w/o 1.4B") average top-1 predictions, over the 17 tasks. are less accurate.

| Scaling law fit | Train set | ARC-E [23] | LAMBADA [77] | OpenBook QA [68] | HellaSwag [126] | 17 eval |
|---|---|---|---|---|---|---|
| Table 1 | C4 [88, 27] | 28.96% | 15.01% | 16.80% | 79.58% | 0.14% |
| Table 1 w/o 1.4B | C4 [88, 27] | 0.92% | 2.04% | 96.16% | 61.79% | 0.42% |
| Table 1 | RedPajama [112] | 5.21% | 14.39% | 8.44% | 25.73% | 0.05% |
| Table 1 w/o 1.4B | RedPajama [112] | 8.13% | 11.07% | 7.56% | 30.98% | 10.64% |
| Table 1 | RefinedWeb [82] | 26.06% | 16.55% | 1.92% | 81.96% | 2.94% |
| Table 1 w/o 1.4B | RefinedWeb [82] | 15.39% | 6.26% | 6.79% | 6.52% | 15.79% |

Table 8: **Token multipliers of existing models.** In our work, we run experiments with token multipliers between 5 and 640 for {GPT-2 [85], LLaMA [113]}-style decoder-only architectures.

| Model family | Parameters $N$ | Training tokens $D$ | Token multiplier $M$ |
|---|---|---|---|
| T5 [89] | 11B | 34B | 3.1 |
| GPT-3 [16] | 175B | 300B | 1.7 |
| Gopher [86] | 280B | 300B | 1.1 |
| Chinchilla [45] | 70B | 1.4T | 20.0 |
| LLaMA [113] | 7B | 1T | 140.0 |
| LLaMA [113] | 70B | 1.4T | 20.0 |
| LLaMA-2 [114] | 7B | 2T | 290.0 |
| LLaMA-2 [114] | 70B | 2T | 30.0 |
| XGen [74] | 7B | 1.5T | 210.0 |
| MPT [110] | 7B | 1T | 140.0 |

