# OpenReview forum: "Language models scale reliably with over-training and on downstream tasks"
_NeurIPS.cc/2024/Conference — Submitted to NeurIPS 2024_

### Official Review · Reviewer_CG6A · 2024-07-08

**Soundness:** 3
**Presentation:** 3
**Contribution:** 2
**Rating:** 7
**Confidence:** 3

**Summary:**

While existing scaling law studies look at compute-optimal pretraining, this paper considers scaling laws in the context of both pretraining and downstream performance. They perform scaling experiments and find that performance is predictable even in overtraining, and average downstream performance is also predictable.

**Strengths:**

I think this is a solid paper that attempts to answer an important question. While I’m concerned about the lack of novelty (see the weaknesses), I overall lean towards accepting the paper. I think that the fact the paper reproduces findings from other papers using a different methodology is a good sign that the overall results are correct, and is valuable information (e.g. [Owen 2024](https://arxiv.org/pdf/2401.04757) also finds that average downstream performance is more predictable than for individual downstream tasks). To me, this is the primary contribution of the paper, and an important one as such.

**Weaknesses:**

To my mind, the largest weakness of this paper is the lack of novelty. In particular, my understanding is that the most important findings are that (1) performance is predictable in overtraining, and (2) average downstream performance is predictable. I’m not sure why (1) should be surprising – doesn’t a parametric scaling law, such as method 3 from the Chinchilla paper, also give the ability to predict loss when overtraining? I think the authors could improve the motivation for this consideration by providing a back of the envelope calculation: for instance, I plugged in the model size and dataset size for Gopher 280B into the Chinchilla scaling law and got a predicted test ppl of ~7.3 on MassiveText. However, Gopher actually had a (validation) perplexity of ~8.1, this constitutes a relative error of around 10% – substantially larger than the relative errors obtained by the authors of this paper. If the authors can provide an argument of this sort I'd find that helpful.

I thought (2) was a more interesting claim, but I’ve seen this analyzed in Owen 2024 (https://arxiv.org/pdf/2401.04757), albeit with a different methodology. As such, I felt that the core results of the paper weren’t very novel. However, I’d be happy to update my assessment if the authors can provide evidence that my understanding is incorrect.

One interesting point that the authors mentioned is that performance on individual tasks is less predictable. But this is only mentioned in passing, and I felt that it could be expanded upon a fair bit. What are the implications of this observation? Are there any patterns for which individual tasks are or aren’t predictable?

I’m slightly concerned about data leakage being an issue for the downstream tasks, given that the training data (The Pile and RedPajama) covers a wide swath of the internet, and some of the downstream benchmarks have been criticized for data leakage or having label errors.

Minor comment: The last paragraph of section 5 is a bit confusing: “There has been a rise in over-trained models [113, 114] and accompanying massive datasets [112, 82, 104, 3]. For example, Chinchilla 70B [45] is trained with a token multiplier of 20, while LLaMA-2 7B [114] uses a token multiplier of 290.” This makes it sound a bit like Chinchilla is overtrained, which I don’t think the authors are trying to say, so I’d suggest something like the following instead: “For example, while Chinchilla 70B is trained compute-optimally with a token multiplier of 20, LLaMA-2 7B…”

**Questions:**

- Doesn’t a standard parametric scaling law fit, such as method 3 from the Chinchilla paper, also allow one to predict loss when overtraining?
- How do the results from this paper differ from those in this other paper, which also looks at downstream performance? https://arxiv.org/pdf/2401.04757
- If I plug in the coefficients from the Chinchilla scaling law into equation 8, with $\alpha = 0.34$ and $\beta = 0.28$, I find that the predicted value for $\eta$ is $\frac{\alpha \beta}{\alpha + \beta} \approx 0.154$. In comparison, the values of $\eta$ in table 6 are generally around 0.25. What’s the cause of this difference, and how do you know?
- Did the authors consider alternative functional forms for the downstream performance scaling law, besides the negative exponential?
- Did the authors check for data leakage? How?

**Limitations:**

I felt that the authors did a good job describing some of the limitations of their work.

---

> ### Author Rebuttal · Authors · 2024-08-07
>
> Thank you for the attention to our work! Please see below for responses to your review. We are happy to provide more clarification or results should it be helpful!
>
> **Over-training novelty.**
> Thank you for pointing out that Chinchilla Approach 3 implies that over-trained model behavior is predictable. We agree, which is why we framed Equation (4) as a reparameterization of Chinchilla Approach 3 rather than as a novel scaling law (L101-109). While Approach 3 implies reliable scaling in the over-trained regime, this is not empirically verified in the Chinchilla paper. Even though the equation implies a phenomenon, it may not be empirically true. Hence, we argue that our over-training results are empirically novel and hence valuable to the community. Furthermore, we consider validation loss, while Chinchilla considers only training loss, and we explicitly measure relative error between predictions and ground truth to actually quantify how good scaling fits are. These features are missing in the original Chinchilla paper; however, we feel they are important for setting scaling research on solid footing.
>
> **Downstream error prediction novelty.**
> Thank you for bringing up the Owen 2024 paper, which we were not aware of at the time of our submission. We added the reference to our main related work section to contextualize that others find a relationship between compute and downstream average error. Our main innovation relative to the Owen study is again empirical. The models considered in the Owen study are not standardized: different architectures, training codebases, optimization schemes, and training datasets––to name a few. Each of these factors introduces confounders. In contrast, we create a standardized, open-source setting, which controls these factors.
>
> **Comparison to Gopher 280B.**
> Thank you for pointing out that the Gopher 280B model does not seem to follow Chinchilla scaling laws. Here we note that Gopher 280B was trained on 300B tokens, which amounts to a token multiplier of $M=\sim1.1$, which is far from the $M=\sim20$ that the Chinchilla team finds is compute optimal on MassiveText. Given that Gopher 280B is under-trained, we might expect its scaling behavior to be less predictable, as also observed in our under-training experiments at $M=5$ (see L252, Appx. Figure 9).
>
> **Predictability on individual tasks.**
> Thank you for mentioning that individual tasks being hard to predict is interesting! We agree and will expend in L236-249. Particularly, we believe that this observation motivates future work on understanding interactions between training sets and downstream eval predictability. Looking at Table 2, it appears that predictably is influenced by training distributions. For example, training on RedPajama allows for predicting relative error for the 7B run on ARC-Easy at $\sim 5\\%$; however, the prediction error is much higher for C4 and RefinedWeb trained models at $>26\\%$. While influence functions [1, 2] are one promising avenue; we believe there is much more work to be done here. Also, looking at the ablations over downstream eval choices in Appendix Figure 8 suggests that generally adding more evals trends towards better predictability.
>
> **Dataset leakage concerns.**
> Thank you for bringing this up, dataset leakage is indeed an important problem to consider and can be hard to mitigate in web-scale data. We used standard, open-source datasets as is and did not conduct additional decontamination past what was done in the original releases. However, there has been recent evidence suggesting that contamination, even when it does exist, may not be catastrophic. For example the DataComp-LM project finds, in their Section 4.6,  that when explicitly decontaminating against MMLU, performance is comparable (51.8 without decontamination and 52.7 with decontamination). We also note that evaluation in NLP is an active area of research and has many open problems, which are not the focus of our study. We hope that as the science of evaluation in NLP advances, researchers will revisit our scaling testbed, critique its shortcomings, and iterate on methodology.
>
> **Related work wording.**
> Thank you for mentioning that it is currently unclear that the Chinchilla 70B model is *not* over-trained. We have applied your suggestion to make this clear.
>
> **Discrepancies in scaling exponents with Chinchilla.**
> Thank you for catching this! After reviewing this discrepancy, we realized that we printed values of $\alpha$ not $\eta$. As mentioned in L108, $\eta = \alpha / 2$. Hence our values for $\eta$ are actually 0.121, 0.136, and 0.127, more closely matching the Chinchilla value. Thanks again for your diligence looking through the Appendix, we have fixed the table.
>
> **Exponential decay relating downstream performance and loss.**
> We did not extensively explore alternatives to exponential decay and rather considered empirical phenomena to inform our choice. L131-132 provides the intuition that the functional form should be bounded from above by $\epsilon$, which represents model performance close to random chance error. The error is naturally bounded from below as loss approaches the irreducible loss $E$.
>
> ---
> **Additional references.**
>
> [1] Pang Wei Koh and Percy Liang. *Understanding Black-box Predictions via Influence Functions.* ICML, 2017. https://arxiv.org/abs/1703.04730.
>
> [2] Roger Grosse, Juhan Bae, Cem Anil, Nelson Elhage, Alex Tamkin, Amirhossein Tajdini, Benoit Steiner, Dustin Li, Esin Durmus,
> Ethan Perez, Evan Hubinger, Kamile Lukosiute, Karina Nguyen, Nicholas Joseph, Sam McCandlish, Jared Kaplan, Samuel R. Bowman. *Studying Large Language Model Generalization with Influence Functions.* arXiv, 2023. https://arxiv.org/abs/2308.03296.

---

> > ### Comment · Reviewer_CG6A · 2024-08-11
> >
> > Thanks for the responses to my questions! I believe this addresses my main concerns. I also think I was being a bit harsh on the novelty point, because even though there's a lot of overlap in results the methodology is quite different and probably more reliable than Owen 2024. As such, I'll increase my score.

---

> > > ### Author Response · Authors · 2024-08-14
> > >
> > > Thanks! We appreciate you reconsidering your score post-rebuttal!

---

### Official Review · Reviewer_RC8r · 2024-07-14

**Soundness:** 2
**Presentation:** 3
**Contribution:** 2
**Rating:** 5
**Confidence:** 3

**Summary:**

The authors propose a scaling law for the “Chinchilla over-trained” regime where models are trained on many more tokens (in this paper, up to 30x) than Chinchilla-optimal. They motivate a scaling law relating pre-training compute and “over-training” to validation loss. They empirically demonstrate that the proposed scaling law accurately predicts the validation loss of a 1.4B 32x over-trained model and a 6.9B Chinchilla-optimal model. They then study a simple scaling law relating perplexity to downstream benchmark error. They select a subset of 17 benchmarks for which a 154M parameter models performs 10% above random chance accuracy, and show show that average downstream error of the 1.4B and 6.9B models is predictable.

**Strengths:**

The setting considered by the paper is very relevant at the moment, as major model releases in the past year fall precisely in the “Chinchilla over-trained” regime. The work is also novel, as I am not aware of prior work that proposes and empirically validates scaling laws tailored for the Chinchilla over-trained regime.

The proposed scaling law in the over-trained regime is well-motivated both by prior scaling laws and by further empirical observations in the over-trained regime (Figure 2). While the models trained (5-8 1e21 FLOPs) are at least two orders of magnitude smaller than the latest open models (e.g., Gemma 2, Llama 3, Qwen 2), the experiments are of a large enough scale for a proof of concept.

The paper is clear and it gives sufficient details on the experimental set-up.

**Weaknesses:**

My main concerns are twofold: authors do not compare with the standard scaling laws of Kaplan et al. (fitted on their own testbed), and it is unclear how authors choose the models used to fit their scaling laws (Table 1).

Authors do not compare their over-trained scaling law with that proposed by Kaplan et al. The authors could fit this scaling law as in Hoffman et al. , Section 3.3, without any additional model training. Specifically, how well can the standard Kaplan et al. law predict the validation loss of the 1.4B over-trained model, when fitted on the model testbed with N < 1B described in Section 3.2?

Regarding the claim that validation loss (resp accuracy) is predictable with 300x (resp. 20x) less compute, I find this misleading, since authors train and evaluate a reasonably large model testbed, but only report the compute required for the 5 (resp. 6) models that they ultimately choose for the fit.  The authors do not discuss how this “train”/“test” split was chosen. Clearly, the train/test split should be chosen before seeing the evaluation results for any of the models, rather than including models until the fit seems "good enough", or choosing the smallest subset for which the fit is “good enough”. Otherwise, both the claim of 300x/20x compute as well as Figure 5 are misleading. Similarly, Figure 1 would be misleading, and it should include all models with N < 1.4B.

The authors consider token multiplier M <= 640, however current models are even more overtrained. If am not mistaken, for Llama 3 8B, M ~= 2000. Demonstrating the validity of the proposed scaling laws for the amount of over-training of current models would have been ideal, even at smaller model scales.

Lastly, the proposed scaling laws are validated at substantially lower compute scales than current models with publicly available weights. It would have been ideal to at least see results for over-trained 7B models. I understand that the experiments presented in the paper already require a substantially amount of compute, and more closely matching the compute scales of recent models would be unfeasible for most research labs — therefore I am not taking this point into consideration when scoring the paper.

**Questions:**

* How well can the standard Kaplan et al. law predict the validation loss of the 1.4B over-trained model, when fitted on the model testbed with N < 1B described in Section 3.2?
* How were the models in Table 1 chosen? Were they chosen prior to the other models being evaluated? Was it apparent that including only one over-trained model, namely 11M, 320M, would suffice?
* Figure 18 shows that HellaSwag and ARC-Easy scores are very correlated with validation loss. However, Table 2 shows very large individual top-1 error for HellaSwag and ARC-Easy. How can this be?

**Limitations:**

Limitations are adequately discussed.

---

> ### Author Rebuttal · Authors · 2024-08-07
>
> **Comparison to Kaplan et al.**
> Thank you for mentioning this important prior work. Unfortunately, the methodology in Kaplan et al., which utilizes early-stopped models and a different learning rate schedule, is not compatible with our scaling testbed, which is similar to the more contemporary Hoffmann et al. setup. Additionally, we argue that Kaplan et al. is not a practically relevant point of comparison given recent work (Porian et al., 2024 [1]) identifying methodological weaknesses in the Kaplan et al. scaling study. Specifically, Porian et al. find errors in the way Kaplan et al. discount last layer compute and set warmup for small models. We chose to base our scaling study on the more recent Chinchilla paper given its wide adoption (e.g., Muenninghoff et al., 2023) and its intentionality towards correcting methodological oversights in Kaplan et al.
>
> **Clarification on experimental setup choices: model configurations in Table 1 and train/test splits.**
> Thank you for bringing this up. We detail our approach to selecting model configurations in Section 3.2 and Figure 4. In brief, we train models in a large grid search on a mixture of Pile and RedPajama data. We attempt to predict the validation loss on OpenLM validation data (the OpenLM codebase, recent arxiv papers, and news articles) for larger 1.4B and 7B chinchilla optimal runs. At this stage we do *not* consider 1) over-training, 2) C4 eval which is used for test loss evals, 3) C4, RedPajama, or RefinedWeb non-mixed datasets, or 4) downstream task performance. Hence our grid search happens in a validation environment, which we are confident is not indexed on our test setup. As for choosing values of $M$, we wanted to test extrapolation to a $N=1.4B, M=640$ over-trained run. Hence, we chose a $M=320$ datapoint and happened to do so at the smallest parameter scale to save compute. To better understand potential human bias that went into this decision, we plot relative error vs. many different potential configurations in Appx. Tables 14-16. The takeaway is that while our eventual choice showed favorable trade-offs between compute and predictive power, there are ways to create more accurate scaling laws with privileged knowledge of test metrics.
>
> **Clarification on reported compute savings for scaling laws.**
> Thank you for the opportunity to clarify our 20x or 300x compute savings for our scaling laws relative to our largest runs. While hyperparameter searches can be expensive, our team bore the upfront cost to find and open-source reliable configurations, which the community can now use for future scaling studies. Hence our claim is that, using our final configurations, one should be able to predict large runs with the aforementioned compute multiples, which we feel is justified.
>
> **Over-training in Llama3 8B.**
> Thanks for bringing up the Llama3 8B release. We note that this release happened after the NeurIPS deadline and that our token multiplier range up to $M=640$ is reasonable for models at the time we submitted. Also Llama3 8B is an outlier in that many popular models are less over-trained (e.g., Mistral 7B). Additionally, it is unclear if, at 15T tokens, Llama3 8B was trained for a single epoch. Practically, training a 1.4B parameter model for $M=\sim 2000$ worth of tokens is prohibitive in our setting as 1) this is more compute than we have access to unfortunately and 2) this requires 2.8T tokens for a single-epoch run, which is larger than public datasets at the time of our training runs.
>
> **Many open-source models are larger than 7B parameters.**
> Thank you for pointing this out and also recognizing we were not able to train Llama-sized models given compute limitations. We agree that verifying scaling trends is valuable for larger models. However, we do not feel that this weakness in our manuscript is a critical flaw for a couple of reasons:
> 1. Increasingly, teams are pushing the capabilities of models with under 7B parameters. For instance, Phi-2 (2.7B parameters) and Gemma 2B (2B parameters) are both performant models.
> 2. One of the main motivations of over-training is to have a model with fewer parameters to save on inference costs (L88-91). Hence, we argue it makes sense to study over-training in a low parameter regime to see how far small models can be pushed with over-training.
>
> This being said, we attempt to address the spirit of your concern by predicting the validation loss of Llama2 7B and 13B models, under the assumption that these models were trained on datasets similar to RefinedWeb. To run this experiment, we re-tokenize RefinedWeb with the Llama2 tokenizer and re-train our small-scale models from Table 1. As we see in the attached pdf (Figure A), with this assumption, our over-trained scaling laws predict accurately both models’ performance. Note, we must make the aforementioned data assumption as scaling laws are fit on a suite of models trained and evaluated on standardized distributions. To truly have a clean experiment, we would need access to the Llama2 training data and internal details. Nevertheless, we feel this experiment suggests over-trained performance can be predictable for larger runs.
>
> **Top-1 error on HellaSwag and ARC-Easy in Table 1 vs. Figure 18.**
> Thanks for bringing this up. Table 2 and Figure 18 show different things. Table 2 shows predictions based on only the six configurations from Table 1. Figure 18, in contrast show the empirical trend with all 104 models we trained. One takeaway here is that it should be possible to achieve reliable downstream prediction on individual tasks with increased compute investment. However, this represents diminishing returns as the scaling law investment approaches the cost of actually training the large scale run.
>
> ---
>
> **New references.**
> [1] Tomer Porian, Mitchell Wortsman, Jenia Jitsev, Ludwig Schmidt, Yair Carmon. *Resolving Discrepancies in Compute-Optimal Scaling of Language Models.* arXiv, 2024. https://arxiv.org/abs/2406.19146.

---

> ### Comment · Reviewer_RC8r · 2024-08-11
>
> Thank you for your response. My two main concerns were not effectively addressed. Having read the other reviewers’ reviews, I’ll increase my score, since I agree that the paper does show that language models scale reliably with over-training.
>
> **Comparison to Kaplan et al.**: What I mean is to consider how well the “standard” functional form used by prior work, which you write in Equation 3, can predict the over-training regime. Without this comparison, it is simply not possible to judge whether the proposed reparametrization in Equation 4 is a valuable contribution in itself or not. Why should practitioners fit Equation 4 and not Equation 3? Note that Equation 3 is the power law of Kaplan (Equation 4.1 in the Kaplan paper) + the irreducible loss term. I included in my review “The authors could fit this scaling law as in Hoffman et al. , Section 3.3, without any additional model training”. Reviewer CG6A echoes a similar weakness “doesn’t a parametric scaling law, such as method 3 from the Chinchilla paper, also give the ability to predict loss when overtraining?”. I fail to understand why you cannot provide this comparison. You do not need to train any additional models, you have the (N, D, L) triplets. It is just a matter of fitting Equation 3.
>
> **Reported compute savings**: I am not referring to the compute spent on the grid search for hyper-parameter tuning. I am referring to the compute required to train the scaling test bed itself, bar the N=1.4B and N=6.9B models which are the ones you ultimately aim to extrapolate to. “Hence, we chose a datapoint M=320 and happened to do so at the smallest parameter scale to save compute”. This is not convincing. A single M=320 datapoint happened to be good enough for the fit. But you did not happen to train a single M=320 model, you trained many models with M > 20. The concern is in choosing the fit/predict split after the models were already trained and evaluated. Otherwise, the stated compute savings and Figure 5 altogether can be misleading.
>
> > we plot relative error vs. many different potential configurations in Appx. Tables 14-16
>
> These plots precisely illustrate my point. Looking at the C4 eval plots, the blue stars are in the Pareto frontier of relative error vs FLOPs spent. The 300x/20x multipliers are not representative of what one would typically obtain, they represent a “best” case scenario, potentially indicative of cherry-picking regarding the fit/predict split.
>
> **Over-training in Llama3 8B and larger models**: I maintain my initial positive assessment that the experiments are of a large enough scale for a proof of concept.

---

> ### Author Response · Authors · 2024-08-14
>
> We apologize for not addressing your concerns; we unfortunately misunderstood your original messages! This said, we appreciate you attending to the other reviews and still reconsidering your score! We aim to address your outstanding concerns below.
>
> **Comparison to Kaplan et al.** There is no reason we cannot provide this comparison. We unfortunately misunderstood your original comment, but please see the table below for the requested results! Note, Equations (3) and (4) in the paper are the same (i.e., both have four free parameters, with the relation given in L108). Concretely, "we reparameterize Equation (3) in terms of compute $C = 6ND$ and a token multiplier $M = D/N$ [to] get, [Equation (4)]" (L107). We include Equation (4) to provide the reader with intuition about what *should* happen when one over-trains (L110-114). We are also careful not to claim Equation (4) as a novel scaling law. For example, in the introduction: "We explain our observations by reparameterizing existing scaling laws in relation to the amount of over-training" (L42-43). Our main contributions with respect to over-training are empirical (as detailed in our response to reviewer CG6A).
>
> | Scaling Form | Model | Relative Error on C4 eval loss|
> |--------------|-------|--------------------------------|
> | $L(N,D) = AN^{-\alpha} + BD^{-\beta}$ (Kaplan et al. like) | open_lm_1b (M=640.0) | 16.1281% |
> | | open_lm_7b (M=20.0) | 17.2352% |
> | $L(C, M) = E + (aM^{\eta} + bM^{-\eta}) C^{-\eta}$ (our reparameterization of Hoffmann et al.) | open_lm_1b (M=640.0) | 0.7103% |
> | | open_lm_7b (M=20.0) | 0.7320% |
>
>
> **Reported compute savings.** Sorry that we misunderstood your comment! We understand your skepticism about our 300x/20x savings claim and ultimately your concern about cherry-picking. We reiterate, using the configurations in Table 1, a practitioner should expect to predict runs similar to our N=1.4B and N=6.9B runs with the claimed compute savings. We will refine the wording to make sure this claim is more precise and clear.
>
> With respect to the entire scaling testbed, we do feel the need to clarify that the results are not cherry-picked. For the sake of transparency, we detail key moments in our development timeline, specifically for the over-training portion of the project. We hope this will provide clarity and help address your very legitimate concerns.
> 1. Chinchilla Approach 3 hints that scaling should be reliable; however, they do not directly measure (in terms of relative error), how good scaling predictions are emperically. Hence, we started our investigation by trying to reproduce Chinchilla. This resulted in a large grid search, as we did not understand how to set hyperparameters to get reliable scaling trends. The grid search ultimately culminated in Section 3.2 of our writeup. Note, at this stage, we had not touched any of our downstream evaluations or our main training distributions. In fact, at this stage we did not even have a project idea! We were, however, largely able to reproduce Chinchilla Approach 3 and observed reliable scaling for compute-optimal models.
> 2. At this stage, we were looking for promising research directions. Motivated by many open-weight releases at the time, we thought it would be nice to understand what happens empirically when one over-trains models. We hypothesized that we should see “parallel lines” in the log-log plot based on the functional form of Approach 3. Hence, we chose a subset of our entire grid search configurations (as detailed in Section 3.2), tokenized C4, and trained these configurations for various token multipliers (i.e., number of tokens). This set of training runs eventually contributed to our scaling testbed. We did in fact see parallel lines and then set out to understand if this phenomenon could be reliably extrapolated with some of our smaller-scale runs.
> 3. At first, we thought we could predict over-trained behavior with only Chinchilla optimal models. However, after playing around with equations and parameterizing with token multipliers, we realized that to extrapolate in M, we would have to fit to at least 1 data point where $M \neq 20$ (i.e., at least one non-compute-optimal run). We knew that we would have enough compute to over-train our largest run at $M=640, N=1.4$B. From our scaling testbed we chose the smallest config ($N=11$M) with the largest token multiplier ($M=320$) such that we could still probe for extrapolation in $N, M$. The other models in Table 1 are, somewhat arbitrarily, Chinchilla optimal.
> 4. We constructed Appx. Tables 14-16 to ablate our decisions. Our meta-analysis here is that the combination of our compute constraints and intuition that small models (e.g., over-trained 11M models) could give reasonable signal were reasonable.

---

### Official Review · Reviewer_7fVK · 2024-07-27

**Soundness:** 3
**Presentation:** 3
**Contribution:** 3
**Rating:** 5
**Confidence:** 3

**Summary:**

This paper investigates the power laws (scaling laws) of neural language models, particularly from the perspective of over-training and the relationship between validation loss (perplexity) and NLP downstream tasks. The authors define over-training as the situation where runs consume large amounts of computational resources, and they introduce a token multiplier, M. It is computed by D / N, where D is the number of training data tokens and N is the number of parameters. Through various model training setups and three different training corpora, the authors demonstrate that the validation loss can be computed and predicted using an equation that includes M. They also introduce another equation that illustrates the relationship between validation loss and downstream task error.

**Strengths:**

The main contribution of this paper is the exploration of the scaling laws in language models concerning over-training and NLP downstream tasks. These results, including equations and practical outcomes, are beneficial for researchers and engineers developing large language models. As highlighted by the authors, these insights are valuable for researchers in their future work.

**Weaknesses:**

The authors mentioned several limitations and future work in the paper. I agree with them, and especially the ‘scaling up’ part is the primary concern of this paper. The model sizes range from 0.011B to 6.9B, but open-source models are larger than these sizes - for instance, Llama 2 starts at 7B, and Llama 3 starts at 8B [1]. Furthermore, model size is crucial for techniques such as CoT [2]. I hope to hear the authors' opinions on this concern.

**Questions:**

- M could be related to overfitting if N is large but D is not sufficient. The authors mentioned that when M equals 5 (under-training), the scaling becomes unreliable. I assume this is because the models are not well trained—underfitting. Could you explain and discuss the relationship between over-training and overfitting?

- (Minor) It is hard to distinguish between M=320 and M=640 in the figures. Could you change the colors to make them more distinguishable?

**Limitations:**

Yes

---

> ### Author Rebuttal · Authors · 2024-08-07
>
> Thank you for the attention to our work! Please see below for responses to your review. We are happy to provide more clarification or results should it be helpful!
>
> **Many open source models are larger than 7B parameters.**
> Thank you for pointing this out––we agree that verifying scaling trends is valuable for larger models. However, we do not feel that this weakness in our manuscript is a critical flaw for a couple of reasons:
> 1) Increasingly, teams are pushing the capabilities of models with under 7B parameters. For instance, Phi-2 (2.7B parameters) and Gemma 2B (2B parameters) are both performant models.
> 2) One of the main motivations of over-training is to have a model with fewer parameters to save on inference costs (L88-91). Hence, we argue it makes sense to study over-training in a low parameter regime to see how far small models can be pushed with over-training.
>
> This being said, we attempt to address the spirit of your concern by predicting the validation loss of Llama2 7B and 13B models, under the assumption that these models were trained on datasets similar to RefinedWeb. To run this experiment, we re-tokenize RefinedWeb with the Llama2 tokenizer and re-train our small-scale models from Table 1. As we see in the attached pdf (Figure A), with this assumption, our over-trained scaling laws predict accurately both models’ performance. Note, we must make the aforementioned data assumption as scaling laws are fit on a suite of models trained and evaluated on standardized distributions. To truly have a clean experiment, we would need access to the Llama2 training data and internal details. Nevertheless, we feel this experiment suggests over-trained performance can be predictable for larger runs.
>
> **Over-training vs. overfitting.**
> Thanks for the opportunity to clarify. While over-training and overfitting are related concepts, they have distinct definitions. Given a Chinchilla optimal allocation of tokens to parameters, over-training refers to training on disproportionately more tokens than parameters (L88-91). Critically, validation loss can still go down with more over-training, as seen in Figure 2. Hence, over-trained models are not necessarily overfit to the data (i.e., overfit models necessarily experience increasing validation loss). Also, in our single epoch training setting, there is no data repetition, so over-fitting is not expected to happen. We agree that the terminology can be confusing, especially given the ubiquity of the term overfitting in machine learning. We will add this explanation to the paper for clarity.
>
> **Underfitting and unreliable scaling.**
> Our empirical observation is that under-trained models (trained on less tokens than is Chinchilla optimal) appear to scale unreliably. We cannot, however, make a stronger claim that all underfit models scale unreliably. For example, in Figure 2, we see that the Chinchilla optimal models ($M=20$) actually underfit the data, as training for $M>20$ decreases validation loss. However, the models at $M=20$ do scale reliably, following a power-law with irreducible error.
>
> **Improved color contrast for Figures.**
> We agree that the contrast could be improved for clarity, thanks for pointing this out. We plan to update Figures 1 and 2, which now use the `plasma` matplotlib color pallet. Please see the attached pdf, Figure B, for a sample.

---

> > ### Comment · Reviewer_7fVK · 2024-08-09
> >
> > Thank you for your responses and the revised figures. Your response have addressed my concerns and clarified my questions. Good luck!

---

> > > ### Author Response · Authors · 2024-08-10
> > >
> > > Thanks! Given that we've addressed the concerns, we were wondering if you might reconsider our score. We understand this decision is entirely at your discretion and are thankful for your attention regardless!

---

### Author Rebuttal · Authors · 2024-08-07

We thank the reviewers for their attention to our work, constructive comments, and positive feedback. Specifically, we are grateful for reviewers highlighting our empirical efforts and strengths of our methodology (RC8r, CG6A). We also appreciate their mentioning the relevance of our scaling study for model development (7fVK, RC8r) especially in light of contemporary research releases where model over-training is commonplace.

We also appreciate the reviewers pointing out room for improvement. In the comments below we address all concerns raised by reviewers, including additional empirical evidence when warranted.

We are happy to continue the discussion during the rebuttal period and provide additional clarification and experiments as needed!

Again, we thank all reviewers for helping us improve our work!

(Please see below for a pdf containing rebuttal Figures A and B mentioned in the comments below)

---

### Decision · Program_Chairs · 2024-09-25

**Decision:**

Reject

**Comment:**

The paper proposes a new LLM scaling law covering over-training regime rather than a compute-optimal regime which is widely used in latest LLM training. It also covers validation loss instead of training loss, and downstream performance as well. The authors claim this significantly saves compute.

The reviewers commonly appreciate that it is an important problem in the field, and the paper provides an useful solution. ("These results, including equations and practical outcomes, are beneficial for researchers and engineers developing large language models.", "The setting considered by the paper is very relevant at the moment", "I think this is a solid paper that attempts to answer an important question."). Reviewer RC8r also appreciate it's cleared written and well presented.

There was a common question around the novelty of the paper if the existing scaling law could already provide the information. The authors provides a clarification that the existing scaling laws actually show much larger error in the over-training regime. There was also a common question if the scaling is only shown on small models. The authors provide a clarification that the current trend tends to focus more on small language models and it could be more optimal for the inference. Most of the other questions raised by the reviewers are addressed by the authors.

Overall, the paper provides an important information for some of the models developed these days which are over-trained SLMs. However, even with the clarifications from the authors, it couldn't provide information about larger models and the studies on the scaling law itself not unique. For those reasons, the authors couldn't convince the reviewers to make the scores to meet the NeurIPS publication bar.